# Efficient Large Language Models: A Survey

**Zhongwei Wan**[*]                                                                                     *wan.512@osu.edu*

**Xin Wang**[*]                                                                                           *wang.15980@osu.edu*

**Che Liu**[†]                                                                                            *che.liu21@imperial.ac.uk*

**Samiul Alam**[*]                                                                                        *alam.140@osu.edu*

**Yu Zheng**[‡]                                                                                           *zhengy30@msu.edu*

**Jiachen Liu**[§]                                                                                        *amberljc@umich.edu*

**Zhongnan Qu**[¶ ‡‡]                                                                                     *znqu@amazon.com*

**Shen Yan**[‖]                                                                                           *shenyan@google.com*

**Yi Zhu**[††]                                                                                            *yi@boson.ai*

**Quanlu Zhang**[**]                                                                                      *quzha@microsoft.com*

**Mosharaf Chowdhury**[§]                                                                                 *mosharaf@umich.edu*

**Mi Zhang**[*]                                                                                           *mizhang.1@osu.edu*

[*]*The Ohio State University*    [†]*Imperial College London*    [‡]*Michigan State University*    [§]*University of Michigan*    [¶]*Amazon AWS AI*    [‖]*Google Research*    [**]*Microsoft Research Asia*    [††]*Boson AI*

**Reviewed on OpenReview:** <https://openreview.net/forum?id=bsCCJHbO8A>

## Abstract

Large Language Models (LLMs) have demonstrated remarkable capabilities in important tasks such as natural language understanding and language generation, and thus have the potential to make a substantial impact on our society. Such capabilities, however, come with the considerable resources they demand, highlighting the strong need to develop effective techniques for addressing their efficiency challenges. In this survey, we provide a systematic and comprehensive review of efficient LLMs research. We organize the literature in a taxonomy consisting of three main categories, covering distinct yet interconnected efficient LLMs topics from model-centric, data-centric, and framework-centric perspective, respectively. We have also created a GitHub repository where we organize the papers featured in this survey at <https://github.com/AIoT-MLSys-Lab/Efficient-LLMs-Survey>. We will actively maintain the repository and incorporate new research as it emerges. We hope our survey can serve as a valuable resource to help researchers and practitioners gain a systematic understanding of efficient LLMs research and inspire them to contribute to this important and exciting field.

---

[‡‡]The work is done outside Amazon.

# 1 Introduction

Large Language Models (LLMs) are a type of advanced AI models designed to understand and generate human languages. Recently, we have witnessed a surge in LLMs including those developed by Open AI (GPT-4 (Achiam et al., 2023) and GPT-3 (Brown et al., 2020)), Meta (LLaMA-3 (Meta, 2024), LLaMA-2 (Touvron et al., 2023b), LLaMA-1 (Touvron et al., 2023a)), and Google (Gemini (Team & Google, 2023), PaLM-2 (Anil et al., 2023), PaLM (Chowdhery et al., 2022), GLaM (Du et al., 2022)) as well as many other models such as BLOOM (Scao et al., 2023), PanGu-$\sum$ (Ren et al., 2023b), and GLM (Zeng et al., 2023). These models have demonstrated remarkable performance across a variety of tasks such as natural language understanding (NLU), language generation, complex reasoning (Yang et al., 2024), and domain-specific tasks related to biomedicine (He et al., 2023; Wan et al., 2023; 2022), law (Eliot, 2021) and code generation (Wei et al., 2022b; Chen et al., 2021b). Such performance breakthroughs can be attributed to their massive scales in model sizes and volumes of training data, as they contain billions or even trillions of parameters while being trained on a gigantic amount of data from diverse sources.

Although LLMs are leading the next wave of AI revolution, their remarkable capabilities come at substantial resource demands (Achiam et al., 2023; Du et al., 2022; Chowdhery et al., 2022; Ren et al., 2023b). Figure 1 illustrates the relationship between model performance and model training time in terms of GPU hours for LLaMA series, where the size of each circle is proportional to the number of model parameters. As shown, although larger models are able to achieve better performance, the amounts of GPU hours used for training them grow exponentially as model sizes scale up. In addition to training, inference also contributes quite significantly to the operational cost of LLMs. Figure 2 depicts the relationship between model performance and inference throughput. Similarly, scaling up the model size enables better performance but comes at the cost of lower inference throughput (higher inference latency), presenting challenges for these models in expanding their reach to a broader customer base and diverse applications in a cost-effective way.

The high resource demands of LLMs highlight the strong need to develop techniques to enhance the efficiency of LLMs. As shown in Figure 2, compared to LLaMA-1-33B, Mistral-7B (Jiang et al., 2023a), which uses grouped-query attention and sliding window attention to speed up inference, achieves comparable performance and much higher throughput. This superiority highlights the feasibility and significance of designing efficiency techniques for LLMs.

The overarching goal of this survey is to provide a holistic view of the technological advances in efficient LLMs. As illustrated in Figure 3, we organize the literature in a taxonomy consisting of three main categories, covering efficient LLMs topics from **model-centric**, **data-centric**, and **framework-centric** perspective, respectively. These three categories cover distinct yet interconnected research topics, collectively providing a systematic and comprehensive review of efficient LLMs research. Specifically,

- **Model-Centric Methods:** Model-centric methods focus on both **algorithm-level** and **system-level** efficient techniques where the model itself is the focal point. With billions or even trillions of parameters, LLMs exhibit distinct characteristics (Wei et al., 2022a) compared to smaller-scale models, necessitating the development of new techniques to enhance their efficiency. In §2, we survey efficient techniques that cover research directions related to model compression, efficient pre-training, efficient fine-tuning, efficient inference, and efficient architecture design.

- **Data-Centric Methods:** In the realm of LLMs, the importance of data is as crucial as that of the model itself. Data-centric methods focus on the role of the quality and structure of data in enhancing the efficiency of LLMs. In §3, we survey efficient techniques that cover research directions related to data selection and prompt engineering.

- **LLM Frameworks**: The advent of LLMs necessitates the development of specialized frameworks to efficiently handle their training, fine-tuning, inference, and serving. While mainstream AI frameworks such as TensorFlow and PyTorch provide the foundations, they lack built-in support for specific optimizations and features crucial for LLMs. In §4, we survey existing frameworks specifically designed for efficient LLMs, covering their unique features, underlying libraries, and specializations.

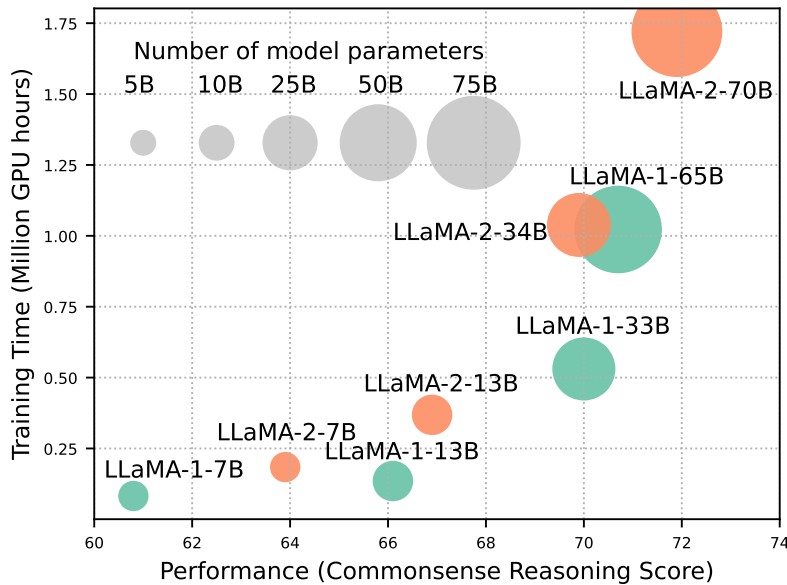

Figure 1: Illustration of model performance and model training time in GPU hours of LLaMA models at different scales. The reported performance is the average score of several commonsense reasoning benchmarks. The training time is based on Nvidia A100 80GB GPU. The size of each circle corresponds to the number of model parameters. The original data can be found in Touvron et al. (2023a;b).

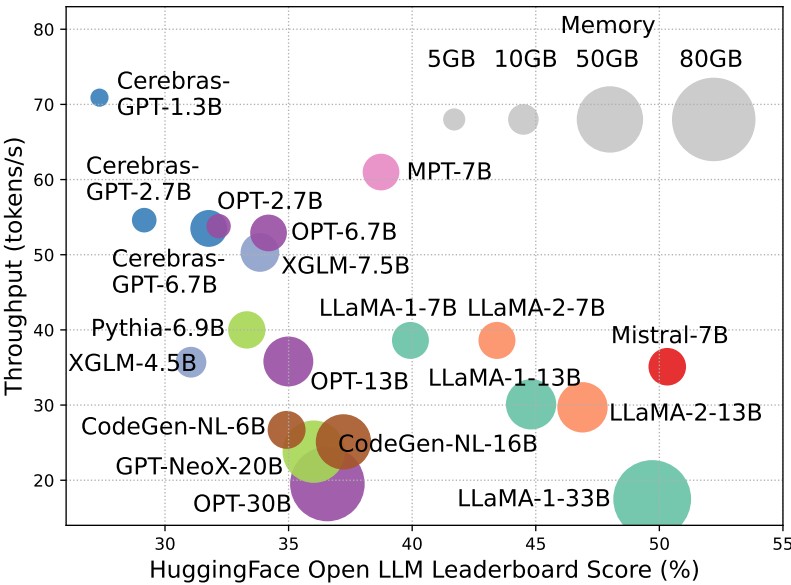

Figure 2: Performance score *vs.* inference throughput for various LLMs. The throughputs are measured on Nvidia A100 80GB GPU with 16-bit floating point quantization. The size of each circle corresponds to the memory footprint (in Gigabytes) of each model when running with a batch size of 1, prompt size of 256, and generating 1000 tokens. The original data can be found in Ilyas Moutawwakil (2023).

In addition to the survey, we have established a **GitHub repository** where we compile the papers featured in this survey at https://github.com/AIoT-MLSys-Lab/Efficient-LLMs-Survey. We will actively maintain it and incorporate new research as it emerges.

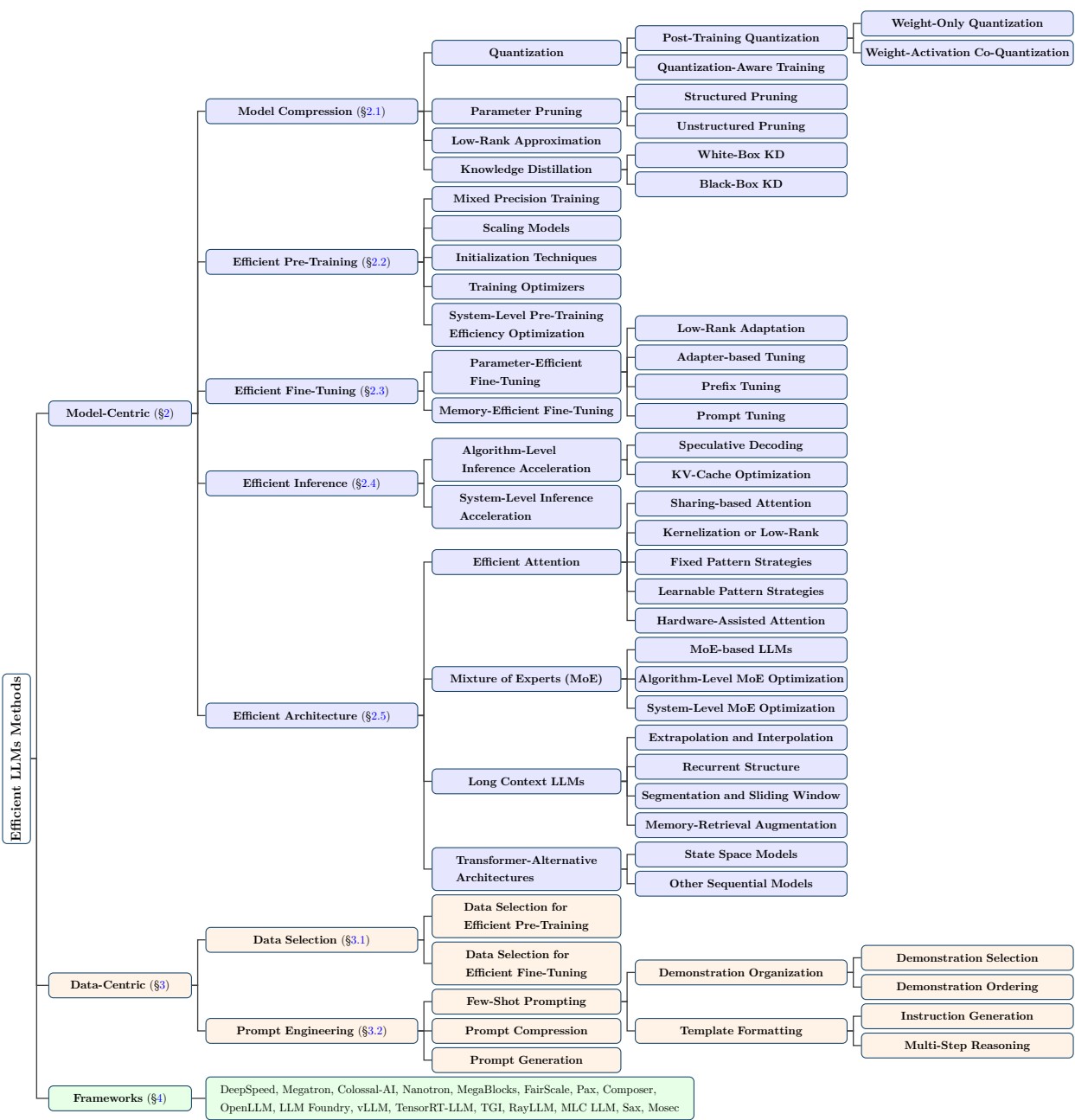

Figure 3: Taxonomy of efficient large language models (LLMs) literature.

Although there are a few surveys on LLMs (Zhao et al., 2023a; Chang et al., 2024; Wang et al., 2023i; Kaddour et al., 2023), this survey provides a focused review and discussion on the literature related to the efficiency aspect of LLMs. There are also surveys on efficient Transformers (Tay et al., 2022) and their training methods (Zhuang et al., 2023). In contrast, this survey specifically focuses on efficiency techniques designed for models of more than billions of parameters. We hope this survey together with the GitHub repository can help researchers and practitioners navigate through the literature and serve as a catalyst for inspiring further research on efficient LLMs.

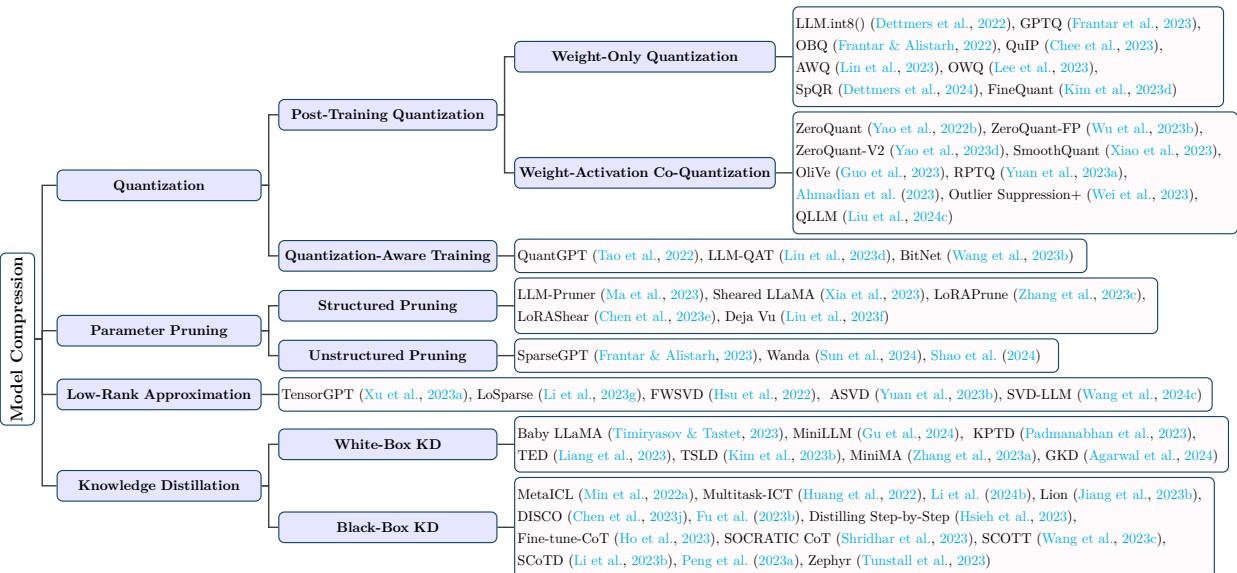

Figure 4: Summary of model compression techniques for LLMs.

## 2 Model-Centric Methods

### 2.1 Model Compression

Model compression enhances efficiency by reducing the sizes and the amount of arithmetic operations of LLMs. Unlike conventional model compression techniques, most LLM compression approaches are designed under the post-training setting to avoid resource-intensive retraining. As summarized in Figure 4, model compression techniques for LLMs can be grouped into four categories: quantization, parameter pruning, low-rank approximation, and knowledge distillation. These four categories are orthogonal to each other, and compress LLMs from different perspectives.

#### 2.1.1 Quantization

Quantization compresses LLMs by converting model weights and/or activations of high-precision data types $\mathbf{X}^{\mathrm{H}}$ such as 32-bit floating point into low-precision data types $\mathbf{X}^{\mathrm{L}}$ such as 8-bit integer (Dettmers et al., 2023) as:

$$\mathbf{X}^{\mathrm{L}} = \mathrm{Round}\left(\frac{\mathrm{absmax}\left(\mathbf{X}^{\mathrm{L}}\right)}{\mathrm{absmax}\left(\mathbf{X}^{\mathrm{H}}\right)}\mathbf{X}^{\mathrm{H}}\right) = \mathrm{Round}\left(\mathcal{K} \cdot \mathbf{X}^{\mathrm{H}}\right), \tag{1}$$

where Round denotes mapping a floating point number into an approximate integer; absmax denotes the absolute maximum of the input elements; and $\mathcal{K}$ denotes the quantization constant. Quantization techniques for LLMs can be classified into post-training quantization (PTQ) and quantization-aware training (QAT). Compared to other LLM compression methods such as parameter pruning and low-rank approximation, quantization methods have been shown to achieve superior compression-accuracy trade-offs (Li et al., 2024c).

**Post-Training Quantization (PTQ).** PTQ quantizes LLMs after the model has been trained. To compensate for the accuracy drop, PTQ uses a small calibration dataset to update the quantized weights and/or activations. PTQ in general can be grouped into two categories: weight-only quantization, and weight-activation co-quantization.

- ***Weight-Only Quantization*** focuses on quantizing model weights only. For example, Dettmers et al. (2022) introduce the first multi-billion-scale 8-bit integers (or INT8) weight quantization method named LLM.int8() that significantly reduces memory usage during inference while being able

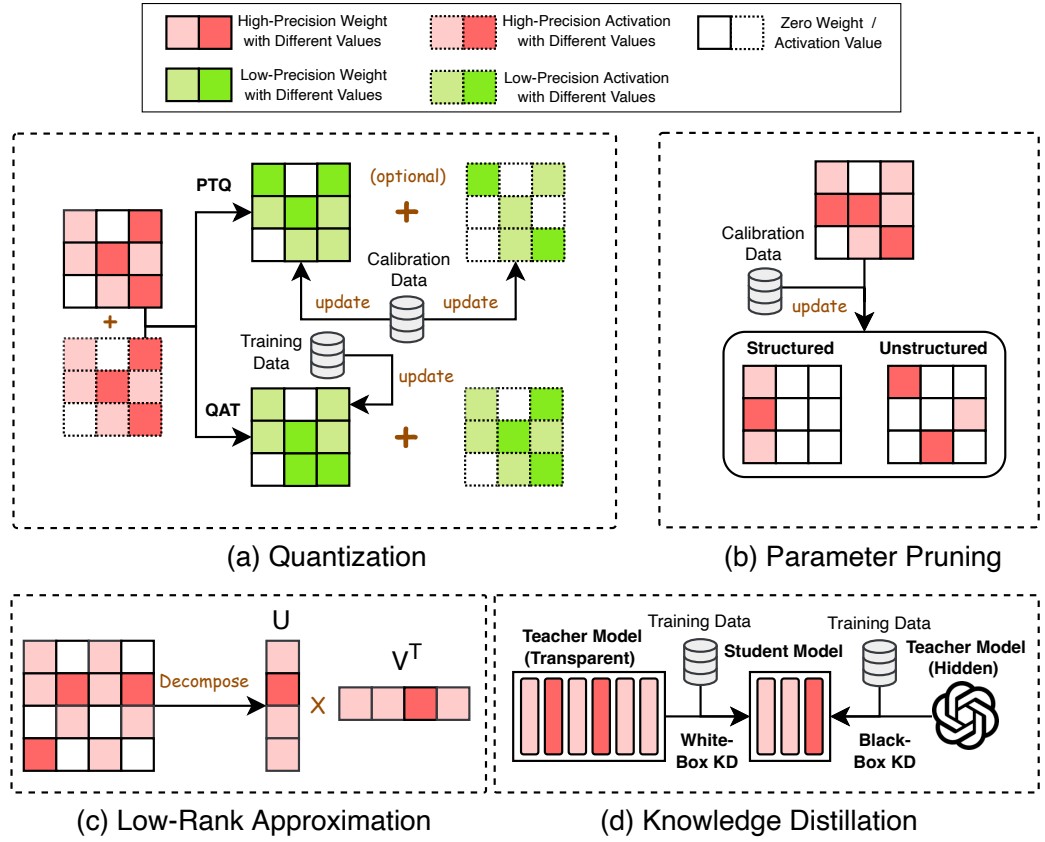

Figure 5: Illustrations of model compression techniques for LLMs.

to maintain the performance of the full-precision model. Frantar et al. (2023) push one step further and propose GPTQ, a post-training weight quantization method that compresses LLM weights to 3 or 4 bits instead of 8 bits. GPTQ employs layer-wise quantization with Optimal Brain Quantization (OBQ) (Frantar & Alistarh, 2022) to update weights with inverse Hessian information. This technique enables quantizing GPT models with 175 billion parameters in roughly four GPU hours with minimal accuracy drop compared to the original model. Furthermore, driven by the insights that quantization can be more effective when model weights and proxy Hessian matrices are incoherent, Chee et al. (2023) propose QuIP, a post-training quantization method that applies incoherence processing to quantize LLMs to 2 bits per weight. As another line of research under weight-only quantization, Lin et al. (2023) observe that there exists a small subset of model weights, characterized by larger activation magnitudes, known as salient weights, play a crucial role in determining the quantization loss. Based on this observation, they propose an approach named activation-aware weight quantization (AWQ) to quantize LLMs while preserving the salient weights in high precision, demonstrating superior performance over GPTQ. Similarly, Lee et al. (2023) observe that activation outliers amplify weight quantization loss. They propose outlier-aware weight quantization (OWQ) to identify those vulnerable weights with activation outliers and allocate high-precision to them. Dettmers et al. (2024) introduce Sparse-Quantized Representation (SpQR) to separate outlier weights that are prone to large quantization errors. These outlier weights are preserved at higher precision, while the remaining weights are compressed to 3-4 bits. Additionally, they introduce a decoding scheme tailored for the SpQR format that enhances the efficiency of inference on a token-by-token basis. Lastly, Kim et al. (2023d) aim to address the issue of outliers that distort the distribution of quantized weights, and propose FineQuant that employs an empirically crafted, heuristic-based approach to allocate varying levels of granularity to different weight matrices.

- ***Weight-Activation Co-Quantization*** differs from weight-only quantization in the sense that it quantizes both model weights and activations. For instance, Yao et al. (2022b) propose ZeroQuant, which combines group-wise quantization for model weights and token-wise quantization for activations. However, ZeroQuant falls short in maintaining accuracy for models with more than 175 billion parameters. To address this issue, Yao et al. (2023d) and Wu et al. (2023b) propose ZeroQuant-FP and ZeroQuant-V2 respectively, both of which utilize low-rank matrices to recover the accuracy drop. A key challenge of weight-activation co-quantization is that due to the existence of outliers, activations are more difficult to quantize than model weights (Bondarenko et al., 2021). To address this challenge, Xiao et al. (2023) propose SmoothQuant which introduces a per-channel scaling transformation that migrates the quantization difficulty from activations to weights to achieve lossless quantization of weights and activations to 8 bits for LLMs up to 530 billion parameters. Guo et al. (2023) pinpoint that outliers are critical in weight-activation co-quantization but their nearby normal values are not. Given that, they propose OliVe, which prunes normal values adjacent to the outliers so that the outliers can be encoded with higher precision. Yuan et al. (2023a) identify the challenge of quantizing activations when different channels have disparate ranges. They propose RPTQ, which groups channels in activations that have similar value ranges and applies uniform quantization parameters to the values in each group. Ahmadian et al. (2023) demonstrate that it is possible to suppress large activation outliers at scales as large as 52B. Given the right optimization choices during pre-training, they can quantize models ranging in size from 410M to 52B with minimal accuracy degradation. Wei et al. (2023) observe that the activation outliers in LLMs are asymmetric and tend to cluster in particular channels. Based on this observation, they propose Outlier Suppression+, which introduces operations that shift and scale channels individually to neutralize asymmetric outliers. Lastly, Liu et al. (2024c) propose QLLM, an adaptive channel reassembly method that tackles activation outliers and utilizes calibration data to offset the information loss incurred from quantization. Experimental result shows that QLLM achieves better compression performance than SmoothQuant and Outlier Suppression+ on LLaMA model family.

**Quantization-Aware Training (QAT).** Different from PTQ, QAT quantizes LLMs during the training process, allowing LLMs to learn quantization-friendly representations. Since QAT requires training using the complete training dataset, it is much more expensive and time consuming than PTQ. Tao et al. (2022) propose QuantGPT, which combines contrastive distillation from a full-precision teacher model and logit distillation to a quantized student model during autoregressive pretraining. QuantGPT achieves $14.4\times$ and $13.4\times$ compression rates on GPT-2 and BART with comparable performance with the full-precision models. LLM-QAT (Liu et al., 2023d) uses data generated by LLMs itself to distill knowledge with the objective of quantizing a student model. Specifically, LLM-QAT retains the original output distribution and is capable of quantizing a model irrespective of its initial training data. Besides quantizing weights and activations, LLM-QAT also quantizes the key-value cache, a crucial step for enhancing throughput and accommodating long sequence dependencies in LLMs. Experimental results show that LLM-QAT achieves better performance over training-free methods especially in low-bit settings. Lastly, BitNet (Wang et al., 2023b) pioneers QAT for 1-bit LLMs. It proposes to use low-precision binary weights and quantized activations while keeping optimizer states and gradients high-precision during training. Experimental results show that compared to FP16 Transformer baselines, BitNet is able to achieve competitive performance while substantially reducing memory footprint and energy consumption.

### 2.1.2 Parameter Pruning

Parameter pruning compresses LLMs by removing redundant or less important model weights. Parameter pruning methods for LLMs can be categorized into structured pruning and unstructured pruning.

**Structured Pruning.** Structured pruning focuses on pruning structured patterns such as groups of consecutive parameters or hierarchical structures such as rows, columns, or sub-blocks of the LLM weight matrices. For instance, LLM-Pruner (Ma et al., 2023) introduces a task-agnostic structured pruning strategy that selectively eliminates non-essential interconnected structures using gradient information. LLM-Pruner utilizes a small amount of data to obtain the weight, parameter, and group importance of the coupled structure for LLaMA (Touvron et al., 2023a), and uses LoRA (Hu et al., 2022) to recover accuracy after pruning,

showing competitive zero-shot performance. Sheared LLaMA (Xia et al., 2023), on the other hand, proposes two techniques to improve the performance of LLM-Pruner. The first technique, named targeted structured pruning, prunes a larger model to a designated target shape by eliminating layers, heads, intermediate and hidden dimensions in an end-to-end manner. The second technique, named dynamic batch loading, dynamically configures the composition of sampled data in each training batch based on losses in various domains. Through these two techniques, Sheared LLaMA is able to prune LLaMA2-7B down to 1.3B parameters, achieving superior compression ratio compared to LLM-Pruner. LoRAPrune (Zhang et al., 2023c) introduces a LoRA-based pruning criterion using LoRA's weights and gradients for importance estimation. By employing an iterative structure pruning process to eliminate excess channels and heads, LoRAPrune achieves better efficiency over LLM-Pruner at 50% compression rate. Lastly, given the input, Deja Vu (Liu et al., 2023f) predicts a small set of attention heads and MLP parameters, referred to as contextual sparsity, yields approximately the same output as the dense model. By exploiting such contextual sparsity, Deja Vu is able to achieve much lower latency compared to FasterTransformer without accuracy drop.

**Unstructured Pruning.** Unstructured pruning, on the other hand, focuses on pruning model weights individually. Compared to structured pruning, unstructured pruning has much more pruning flexibility and thus enjoys a lower accuracy drop. However, unstructured pruning incurs irregular sparsification, which in general makes the resulting pruned models difficult to be deployed on hardware except specific types of hardware such as Nvidia Ampere GPUs (Busato & Pool, 2020). For instance, Frantar & Alistarh (2023) introduce SparseGPT, an one-shot LLM unstructured pruning approach that does not require retraining. SparseGPT formulates pruning as a sparse regression problem and solves it by utilizing an approximate solver based on the inversion of the Hessian matrix. In doing so, SparseGPT reaches about 60% unstructured sparsity on models such as OPT-135B while experiencing only a slight performance drop. Sun et al. (2024) propose Wanda, which prunes weights based on the product values of weight magnitudes and their respective input activations. Compared to SparseGPT, Wanda neither relies on second-order information nor necessitates weight update, and is able to achieve competitive performance. Shao et al. (2024) improve the performance of SparseGPT in another way. Specifically, instead of performing the unstructured pruning with a unified ratio for every layer, they propose to utilize Hessian sensitivity-aware mixed sparsity pruning to achieve a minimum of 50% sparsity in LLMs without retraining. This method adaptively assigns sparsity based on sensitivity to minimize the error induced by pruning while preserving the overall level of sparsity.

### 2.1.3 Low-Rank Approximation

Low-rank approximation compresses LLMs by approximating the LLM weight matrix $\mathbf{W}^{m \times n}$ with smaller low-rank matrices $\mathbf{U}$ and $\mathbf{V}$ such that $\mathbf{W} \approx \mathbf{U}\mathbf{V}^\top$, where $\mathbf{U} \in \mathbb{R}^{m \times r}$, $\mathbf{V} \in \mathbb{R}^{n \times r}$, and $r$ is typically much smaller than $m, n$. In doing so, low-rank approximation reduces the number of parameters and enhances efficiency. For example, Xu et al. (2023a) introduce TensorGPT which compresses the embedding layers of LLMs using Tensor-Train Decomposition (TTD). It transforms and breaks down each token embedding and creates an efficient embedding format named Matrix Product State (MPS) that can be efficiently computed in a distributed manner. LoSparse (Li et al., 2023g) improves the performance of TensorGPT by compressing the coherent and expressive components within neurons through low-rank approximation while eliminating the incoherent and non-expressive elements through pruning. As another line of research, FWSVD (Hsu et al., 2022) compresses the weight matrix of an LLM instead of the token embedding matrix via low-rank approximation. Specifically, instead of using vanilla singular value decomposition (SVD), FWSVD proposes a weighted SVD approach which uses Fisher information to weigh the importance of the weights for compression. While FWSVD demonstrates competitive compression results under low compression ratios, it requires calculating the gradients based on the training dataset of the target task to estimate the importance scores, which is task-specific and demands significant computation resources. In contrast, ASVD (Yuan et al., 2023b) proposes a training-free SVD-based approach. It scales the weight matrix based on the activation distribution that enhances the decomposition accuracy and efficiency for model compression. However, neither FWSVD nor ASVD directly correlate singular values with compression loss. Consequently, truncating the smaller singular values might result in increased compression loss. SVD-LLM (Wang et al., 2024c) addresses this drawback by incorporating a truncation-aware data whitening strategy that establishes a direct mapping between singular values and compression loss. Experimental results demonstrate the superiority of SVD-LLM over FWSVD and ASVD in terms of compression performance and speed.

### 2.1.4  Knowledge Distillation

Knowledge Distillation (KD) compresses LLMs by transferring knowledge from a large teacher LLM to a smaller student LLM. Though effective, compared to other LLM compression methods, knowledge distillation methods incur a resource-demanding distillation process. In general, KD for LLMs can be categorized into white-box KD methods and black-box KD methods.

**White-Box Knowledge Distillation.**  White-box KD refers to KD techniques where the parameters or logits of the teacher LLM are used in the distillation process (Gou et al., 2021). For example, as a pioneering effort in this direction, Baby LLaMA (Timiryasov & Tastet, 2023) trains an ensemble of GPT-2 and a collection of smaller LLaMA models using the BabyLM dataset of 10M words. This ensemble is then distilled into a compact LLaMA model with 58 million parameters, which outperforms both its original teacher models as well as a comparable model that was trained without the use of distillation. Gu et al. (2024) observe that conventional KD objectives, such as Kullback-Leibler divergence (KLD), may not be suited for open text generation tasks due to their complex output spaces compared to classification tasks. To address this issue, they propose MiniLLM which minimizes reverse KLD using the gradient of the objective function through policy gradient techniques (Sutton et al., 1999). Experimental results show that MiniLLM achieves better accuracy than conventional KD or directly fine-tuning student models. KPTD (Padmanabhan et al., 2023) demonstrates that white-box KD can transfer and disseminate knowledge from entity definitions into the parameters of a pre-trained language model. Specifically, KPTD creates a transfer set by prompting the language model to generate text based on the definition of the entity. The model parameters are then updated to align the distribution of the student model with that of the teacher model. TED (Liang et al., 2023) introduces a technique for layer-specific task distillation. It uses specially designed filters to align the internal states of both student and teacher models in each layer. These filters extract relevant knowledge from the internal states that is beneficial for the specific task. TED shows considerable and steady gains in performance on both continual pre-training and fine-tuning. TSLD (Kim et al., 2023b) leverages token-level distillation to enhance QAT. It addresses the limitations of layer-to-layer KD in token prediction recovery by reforming intermediate representation and has successfully applied QAT to LLMs. MiniMA (Zhang et al., 2023a) proposes a viewport towards the capacity gap in distilling LLMs, converting it into a principle through analysis and introducing a 3B Language Model that sets a new benchmark for compute-performance pareto frontier. Experimental results show that MiniMA achieves the best accuracy compared to other 3B distilled LLMs. Lastly, Generalized knowledge distillation (GKD) (Agarwal et al., 2024) addresses the issue of distribution mismatch by drawing output sequences from the student model during training. GKD can be applied in combination with other distillation methods to improve their compression performance.

**Black-Box Knowledge Distillation.**  Different from white-box KD, in black-box KD, only the outputs generated from the teacher LLM are used in the distillation process. Inspired by ICT (Chen et al., 2022c) and MetaICL (Min et al., 2022a), where the language model is meta-trained under a wide range of tasks using in-context learning objectives and then fine-tuned for unseen tasks through in-context learning, Multitask-ICT (Huang et al., 2022) introduces a concept known as in-context learning distillation to transfer the few-shot learning capabilities from the teacher model to the student model. Experimental results show that under Multitask-ICT, in-context learning objectives achieve the best performance when combined with language modeling objectives. Similarly, Li et al. (2024b) introduce a hybrid prompting technique that employs multi-task learning along with explanations generated by GPT-3 text-davinci-002 version (OpenAI, 2023). This method is used to distill explanations into smaller models, achieving consistent and significant improvements over strong single-task fine-tuning benchmarks in different scenarios. Experiments on multiple reasoning tasks show that this method even perform better than finetuning or prompting a 60x larger GPT-3 (175B) model by up to 9.5% in accuracy. Lion (Jiang et al., 2023b) introduces an adversarial distillation architecture aimed at enhancing the efficiency of knowledge transfer by incrementally improving the skill level of the student model. Specifically, it prompts LLMs to recognize challenging instructions and creates new complex instructions for the student model, thereby establishing a three-phase adversarial cycle involving imitation, discrimination, and generation. Experimental results show that Lion-13B not only achieves comparable open-ended generation capabilities to ChatGPT but surpasses conventional instruction-tuned models. DISCO (Chen et al., 2023j) prompts a general LLM to produce phrasal perturbations. These generated perturbations are then filtered by a specialized teacher model to distill high-quality counterfactual

Table 1: Pre-training costs of representative LLMs.

| Model | Parameter Size | Data Scale | GPUs Cost | Training Time |
|---|---|---|---|---|
| GPT-3 (Brown et al., 2020) | 175B | 300B tokens | - | - |
| GPT-NeoX-20B (Black et al., 2022) | 20B | 825GB corpus | 96 A100-40G | - |
| OPT (Zhang et al., 2022a) | 175B | 180B tokens | 992 A100-80G | - |
| BLOOM (Scao et al., 2023) | 176B | 366B tokens | 384 A100-80G | 105 days |
| GLM (Zeng et al., 2023) | 130B | 400B tokens | 786 A100-40G | 60 days |
| LLaMA (Touvron et al., 2023a) | 65B | 1.4T tokens | 2048 A100-80G | 21 days |
| LLaMA-2 (Touvron et al., 2023b) | 70B | 2T tokens | A100-80G | 71,680 GPU days |
| Gopher (Rae et al., 2022) | 280B | 300B tokens | 1024 A100 | 13.4 days |
| LaMDA (Thoppilan et al., 2022) | 137B | 768B tokens | 1024 TPU-v3 | 57.7 days |
| GLaM (Du et al., 2022) | 1200B | 280B tokens | 1024 TPU-v4 | 574 hours |
| PanGu-$\alpha$ (Zeng et al., 2021) | 13B | 1.1TB corpus | 2048 Ascend 910 | - |
| PanGu-$\sum$ (Ren et al., 2023b) | 1085B | 329B tokens | 512 Ascend 910 | 100 days |
| PaLM (Chowdhery et al., 2022) | 540B | 780B tokens | 6144 TPU-v4 | - |
| PaLM-2 (Anil et al., 2023) | - | 3.6T tokens | TPUv4 | - |
| WeLM (Su et al., 2023a) | 10B | 300B tokens | 128 A100-40G | 24 days |
| Flan-PaLM (Chung et al., 2022) | 540B | - | 512 TPU-v4 | 37 hours |
| AlexaTM (Soltan et al., 2022) | 20B | 1.3 tokens | 128 A100 | 120 days |
| Codegeex (Zheng et al., 2023) | 13B | 850 tokens | 1536 Ascend 910 | 60 days |
| MPT-7B (Team, 2023) | 7B | 1T tokens | - | - |

data into smaller student models, allowing the smaller models to learn causal representations more reliably. As another line of research, some studies have shown that chain-of-thought (CoT) prompting can elicit language models to solve complex reasoning tasks step by step, with the aim to transfer such ability from large models into smaller ones through black-box KD. For example, to enhance the CoT math reasoning capabilities of smaller models, Fu et al. (2023b) propose a method for instruct-tuning a student model (FlanT5) by distilling the reasoning pathways found in the GSM8K dataset from a teacher model (GPT-3.5 code-davinci-002 (Chen et al., 2021b)). Fine-tuning and distilling smaller models require substantial amounts of training data to match the performance of the large model. To address this issue, Hsieh et al. (2023) propose Distilling Step-by-Step, a technique that uses CoT prompting to extract LLM rationales for extra guidance in training smaller models within a multi-task setting. Experimental results show that Distilling Step-by-Step achieves better performance with much fewer labeled or unlabeled training examples compared to both fine-tuning and standard distillation. Fine-tune-CoT (Ho et al., 2023) utilizes existing zero-shot CoT prompting techniques (Kojima et al., 2022) to create rationales from LLMs. These rationales are then used to fine-tune smaller student models. It also introduces diverse reasoning, a method that employs stochastic sampling to generate a variety of reasoning solutions from teacher models, which serves to enrich the training data for the student models. SOCRATIC CoT (Shridhar et al., 2023) breaks down the original problem into a series of smaller sub-problems and utilizes this decomposition to direct the intermediate steps of reasoning. It is used to train a pair of smaller, distilled models: one specializes in dissecting the problem and the other focuses on solving these sub-problems. SOCRATIC COT is shown to be an effective alternative to CoT, enabling a much smaller model (GPT-2 large) to outperform a 10x larger model (GPT-3 6B). SCOTT (Wang et al., 2023c) uses rationales generated by LLMs to train a student model under a counterfactual reasoning framework. It ensures that the student model does not overlook the provided rationales, thereby preventing it from making inconsistent predictions. Experimental results show that SCOTT can generate CoT rationales that are more faithful than original CoT prompting. Li et al. (2023b) present a method called symbolic CoT distillation (SCoTD) that draws CoT rationales from a LLM using unlabeled data instances. A smaller model is then trained to predict both the sampled rationales and the associated labels. Lastly, Peng et al. (2023a) utilize GPT-4 as a teacher model to generate English and Chinese instruction-based datasets to refine student LLMs. They show that the 52K data points generated by GPT-4 are able to improve zero-shot performance compared to instruction-following data generated from previous state-of-the-art models.

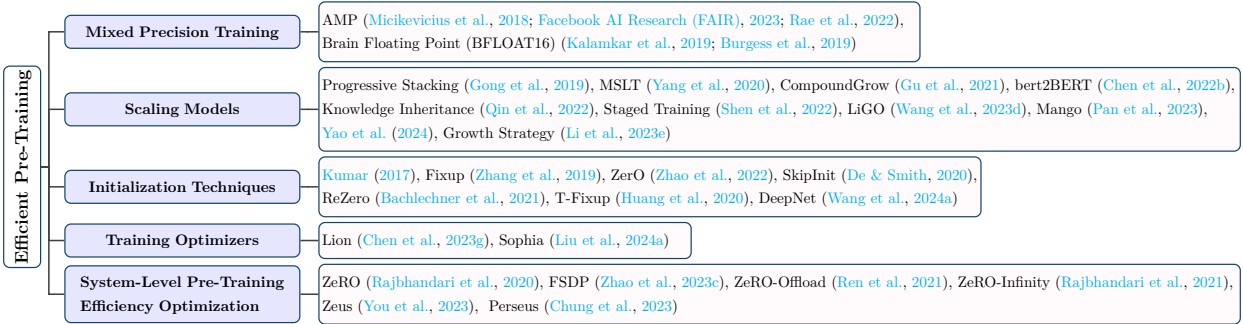

Figure 6: Summary of efficient pre-training techniques for LLMs.

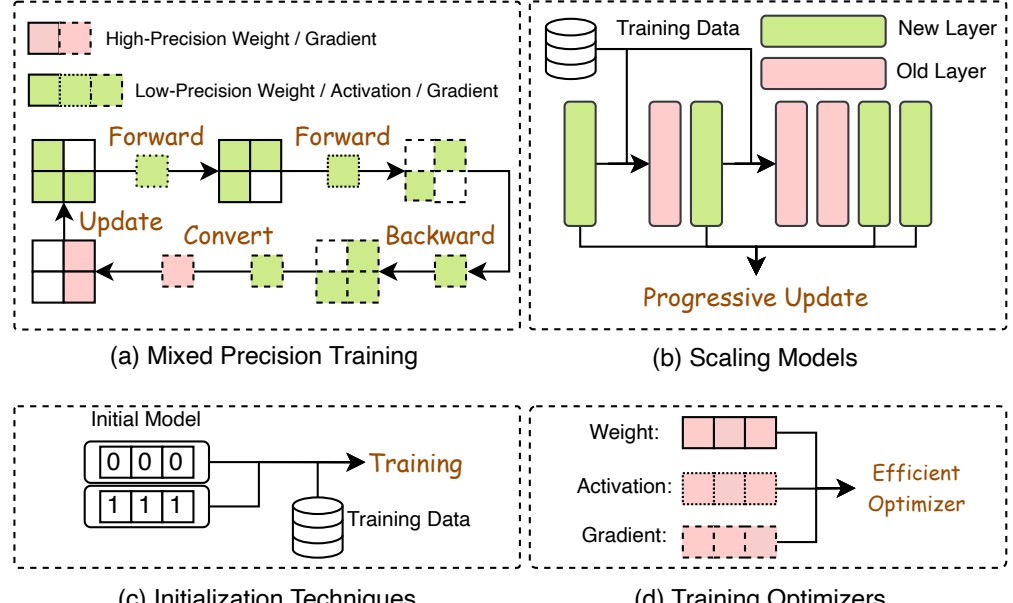

(a) Mixed Precision Training

(b) Scaling Models

(c) Initialization Techniques

(d) Training Optimizers

Figure 7: Illustrations of efficient pre-training techniques for LLMs.

## 2.2 Efficient Pre-Training

As shown in Table 1, pre-training LLMs incurs significant costs. Efficient pre-training techniques focus on reducing the costs of the LLM pre-training process in terms of compute resources, training time, memory and energy consumption. As summarized in Figure 6, enhancing the efficiency of pre-training can be achieved through different and complementary techniques, including mixed precision acceleration, scaling models, initialization techniques, training optimizers, and system-level pre-training efficiency optimization.

**Mixed Precision Training.** Mixed precision training enhances pre-training efficiency by using low-precision models for forward and backward propagation and then converting the calculated low-precision gradients to high-precision ones for updating the original high-precision weights. For example, Micikevicius et al. (2018) propose Automatic Mixed Precision (AMP) to keep a master copy of weights in full-precision (FP32) for updates, whereas weights, activations, and gradients are stored in FP16 for arithmetic operations. Notably, the improved version of AMP (Facebook AI Research (FAIR), 2023) has eliminated the copy of FP32 weights, but the optimizer (AdamW) still uses FP32 internally. Meanwhile, Rae et al. (2022) demonstrate that FP16 in AMP results in accuracy loss due to the restricted numerical range. To address this issue, Brain Floating Point (BFLOAT16), which has a greater dynamic range — i.e., number of exponent bits — than FP16, was proposed (Kalamkar et al., 2019; Burgess et al., 2019) to achieve better training performance.

**Scaling Models.** Techniques based on scaling models accelerate pre-training convergence and reduce training costs by leveraging the weights of a smaller model to upscale to a larger one. For example, Gong et al. (2019) introduce a technique named progressive stacking to transfer knowledge from a simpler model to a more complex one to enhance model training efficiency. Meanwhile, Yang et al. (2020) observe that as the depth of the model increases through progressive stacking, the training speed however decreases. To address this issue, they propose multi-stage layer training (MSLT), which only updates the output and newly introduced top encoder layers while keeping the previously trained layers unchanged. Once all the layers have been trained, MSLT fine-tunes the entire model by updating each layer with 20% of the total steps, making it more time-efficient than the traditional progressive stacking approach. Similarly, Gu et al. (2021) introduce CompoundGrow, which begins with training a small model and then incrementally expands it using a mix of model growth techniques, including increasing input length, model breadth and depth, leading to an acceleration in the pre-training process in wall-clock time compared to progressive stacking. Chen et al. (2022b) propose bert2BERT, which applies function-preserving initialization (FPI) and advanced knowledge initialization (AKI) to transfer the knowledge of a smaller pre-trained model to a large model to improve the pre-training efficiency of the large model. Specifically, FPI enforces the initialized larger model to closely mirror the behavior of the smaller model, laying a strong basis for later optimization; and AKI promotes faster convergence by replicating weights from higher layers. Experimental results show that bert2BERT is able to save a significant amount of training cost over MSLT. Qin et al. (2022) propose Knowledge Inheritance which employs knowledge distillation as an auxiliary supervision during pre-training. This facilitates training a larger model from a smaller teacher model, thereby enhancing both the pre-training speed and the generalization ability. Shen et al. (2022) introduce Staged Training that begins with a small model and progressively increases its depth and breadth through a growth operator. By starting each stage with the results from the previous one, it effectively reuses computation, leading to a more efficient training process compared to previous techniques like CompoundGrow and progressive stacking. Wang et al. (2023d) propose Linear Growth Operator (LiGO) that linearly maps the parameters of a smaller model to initiate a larger one. By using a composition of width-and depth-growth operators further enhanced with Kronecker factorization to capture architectural knowledge, LiGO outperforms bert2BERT which saves about 30% computational costs. Pan et al. (2023) introduce a technique named Mango which establishes a linear relationship between each weight of the target model and all weights of the pretrained model to boost acceleration capabilities. It also employs multi-linear operators to decrease computational and spatial complexity during pre-training, achieving 59.9% acceleration ratio compared to Chen et al. (2022b) and LiGO. Drawing from these scaling techniques and the progressive pre-training (Yao et al., 2024), recent LLMs like FLM-101B (Li et al., 2023e) introduce a growth strategy to cut LLM training costs by expanding model structures offline and resuming from the previous stage's smaller model checkpoint.

**Initialization Techniques.** Initialization plays a key role in enhancing the efficiency of LLM pre-training because a good initialization can accelerate the convergence of the model. Most LLMs employ initialization techniques that were adopted in training smaller-scale models. For example, initialization method introduced by Kumar (2017) balances input and output variances. Fixup (Zhang et al., 2019) and ZerO (Zhao et al., 2022) set the backbone to zero, preserving signal identity. SkipInit (De & Smith, 2020) substitutes batch normalization with a zero-value multiplier. ReZero (Bachlechner et al., 2021) adds zero-valued parameters to maintain identity which leads to faster convergence. T-Fixup (Huang et al., 2020) follows Fixup to adopt rescaling schemes for the initialization of the residual blocks of Transformer models. DeepNet (Wang et al., 2024a) adjusts the residual connection in deep Transformers using Post-LN-init, ensuring stable inputs to layer normalization and mitigating gradient vanishing for stable optimization.

**Training Optimizers.** Popular LLMs such as GPT-3 (Brown et al., 2020), OPT (Zhang et al., 2022a), BLOOM (Scao et al., 2023), and Chinchilla (Hoffmann et al., 2022) are predominately pre-trained using Adam (Kingma & Ba, 2017) or AdamW (Loshchilov & Hutter, 2019) as optimizers. However, both Adam and AdamW are memory hungry and computationally expensive. Some studies (Chen et al., 2023g; Liu et al., 2024a) propose new optimizers to accelerate LLM pre-training. Specifically, Chen et al. (2023g) propose to leverage search techniques to traverse a large and sparse program space to discover optimizers for model training. The discovered optimizer, named Lion (EvoLved Sign Momentum), is more memory-efficient than Adam as it only keeps track of the momentum. Liu et al. (2024a), on the other hand, propose Sophia as a lightweight second-order optimizer that outpaces Adam with doubling the pre-training speed. Sophia

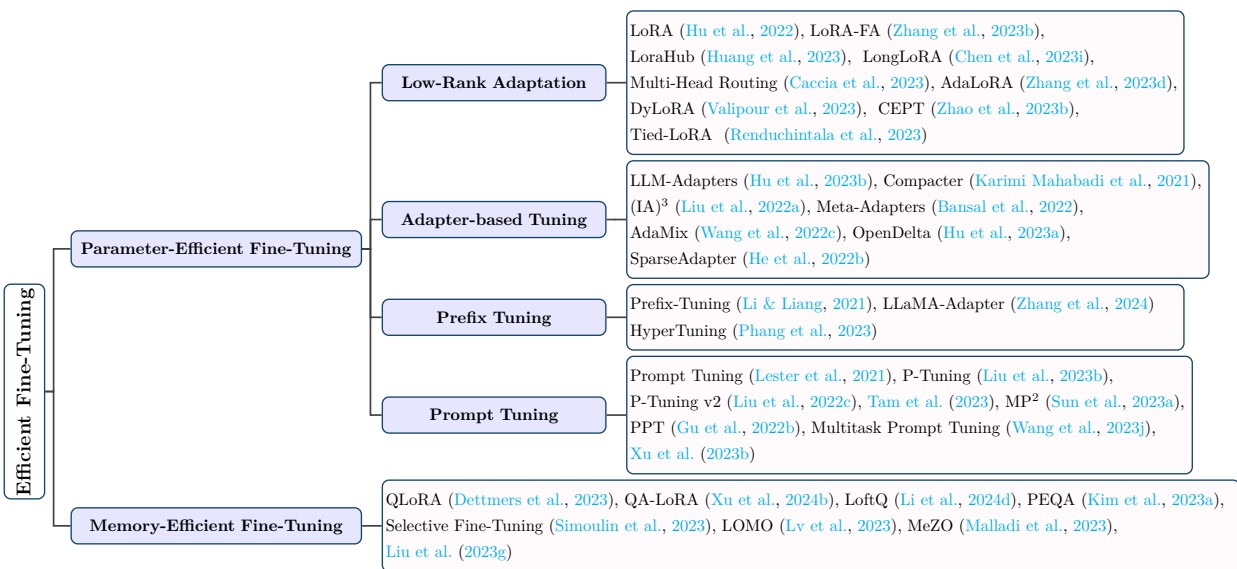

Figure 8: Summary of efficient fine-tuning methods for LLMs.

calculates the moving average of gradients and the estimated Hessian, dividing the former by the latter and applying element-wise clipping. It effectively moderates update sizes, addresses non-convexity and rapid hessian changes, enhancing both memory utilization and efficiency.

**System-Level Pre-Training Efficiency Optimization.** Due to high demand on memory and compute resources, LLMs are usually pre-trained across multiple compute nodes in a distributed manner. Therefore, most system-level optimization techniques are designed in the setting of large-scale distributed training. For instance, Zero Redundancy Data Parallelism (ZeRO) (Rajbhandari et al., 2020) provides three stages of optimization to partition various training states across different devices. Specifically, ZeRO-1 only partitions the optimizer states, whereas ZeRO-2 partitions both the optimizer states and the gradients. ZeRO-3 further partitions the model parameters across devices compared with ZeRO-1 and ZeRO-2. Although runtime memory is further reduced through ZeRO-3, there is about 50% increase in communication volume. Therefore, it is recommended to use ZeRO-3 within a node to minimize the communication time while using ZeRO-1 and ZeRO-2 across nodes. Fully Sharded Data Parallel (FSDP) (Zhao et al., 2023c) shares a similar idea for optimization, and designs a hybrid sharding strategy to allow users to define which nodes or processes to partition the gradients, model parameters, and optimizer states across different nodes. In the case when the weight memory exceeds the aggregated memory that can be provided by all of the compute nodes, ZeRO-Offload (Ren et al., 2021) enables offloading any stage of ZeRO to CPU memory, whereas ZeRO-Infinity (Rajbhandari et al., 2021) provides a mechanism to offload to NVMe drives in addition to CPU memory. However, it is quite difficult to maintain performance using these two alternatives, as the data movement between CPU and GPU is slow. Lastly, training LLMs on numerous GPUs consumes a massive amount of energy, Zeus (You et al., 2023) and Perseus (Chung et al., 2023) are proposed to optimize energy consumption by finding the best GPU-level configurations based on the unique LLM characteristics. The evaluation shows that Perseus reduces energy consumption of large model training by up to 30%.

## 2.3 Efficient Fine-Tuning

Efficient fine-tuning techniques focus on reducing the costs of the LLM fine-tuning process. As summarized in Figure 8, efficient fine-tuning techniques can be grouped into parameter-efficient fine-tuning (PEFT) and memory-efficient fine-tuning (MEFT).

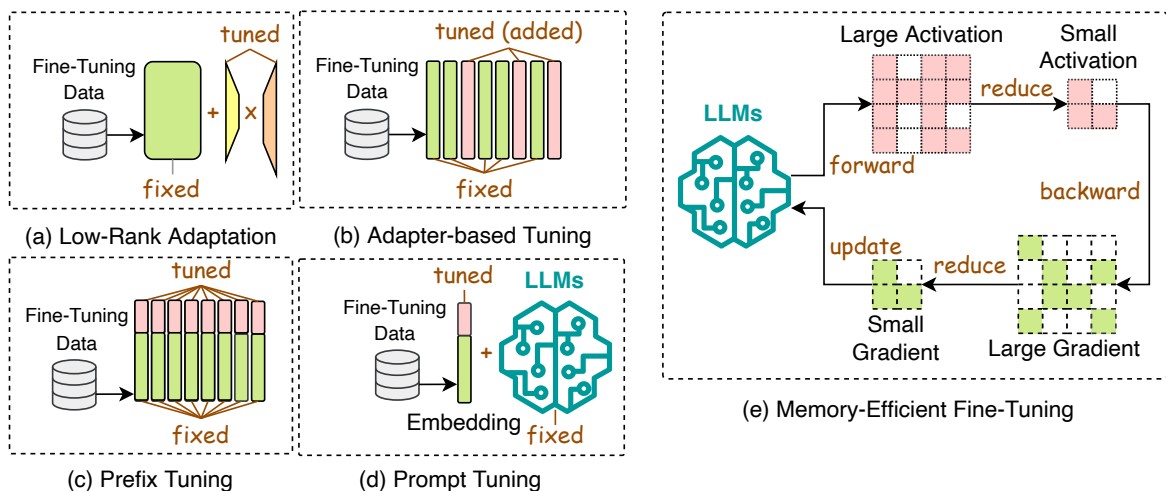

Figure 9: Illustrations of parameter-efficient fine-tuning (a)-(d) and memory-efficient fine-tuning (e).

### 2.3.1  Parameter-Efficient Fine-Tuning

Parameter-efficient fine-tuning (PEFT) adapts an LLM to downstream tasks by freezing the whole LLM backbone and only updating a small set of newly added extra parameters. In general, PEFT methods can be grouped into four categories: low-rank adaptation, adapter-based tuning, prefix tuning, and prompt tuning.

**Low-Rank Adaptation.** Low-rank adaptation (LoRA) (Hu et al., 2022) is a widely used PEFT approach for LLMs. The hypothesis is that the change in weights during model adaptation has a low "intrinsic rank". Hence, LoRA introduces two trainable low-rank matrices $\mathbf{A} \in \mathbb{R}^{m \times r}$ and $\mathbf{B} \in \mathbb{R}^{r \times n}$ and adjusts the weight matrix by $\mathbf{W} \leftarrow \mathbf{W} + \Delta\mathbf{W} = \mathbf{W} + \mathbf{A} \cdot \mathbf{B}$. As such, only the small matrices $\mathbf{A}$ and $\mathbf{B}$ are updated during fine-tuning, while the original large weight matrix remains frozen, making the fine-tuning process more efficient. To enhance the efficiency of LoRA, LoRA-FA (Zhang et al., 2023b) keeps the projection-down weights of $\mathbf{A}$ fixed while only updating the projection-up weights of $\mathbf{B}$ in each LoRA adapter so that the weight modifications during fine-tuning are confined to a low-rank space, thereby eliminating the need to store the full-rank input activations. It achieves comparable accuracy related to full parameter fine-tuning and LoRA. Building on top of LoRA, LoraHub (Huang et al., 2023) explores the composability of LoRA for the purpose of generalizing across different tasks. It combines LoRA modules that have been trained on various tasks with the goal of attaining good performance on tasks that have not been seen before. LongLoRA (Chen et al., 2023i), on the other hand, extends LoRA to the long-context fine-tuning scenario. It introduces shift short attention (S²-Attn), which effectively facilitates context expansion, showing that LoRA is effective for long context when utilizing trainable embedding and normalization. Multi-Head Routing (MHR) (Caccia et al., 2023) extends LoRA to Mixture-of-Experts (MoE) architectures. It outperforms Polytropon (Ponti et al., 2023) when operating with a similar parameter allocation. Notably, it achieves competitive performance while focusing on fine-tuning the routing function alone, without making adjustments to the adapters, demonstrating remarkable parameter efficiency. Zhang et al. (2023d) observe that many PEFT techniques neglect the differing significance of various weight parameters. To address this, they propose AdaLoRA which employs singular value decomposition to parameterize incremental updates and adaptively distributes the parameter budget based on the importance score of each weight matrix. The rank in LoRA is static and cannot be adaptively adjusted during fine-tuning. Valipour et al. (2023) propose DyLoRA to introduce a dynamic low-rank adaptation method that trains LoRA blocks across multiple ranks rather than just one by organizing the representations learned by the adapter module based on their ranks. Different from the above-mentioned methods that apply LoRA-based methods to full-size LLMs, CEPT (Zhao et al., 2023b) introduces a framework that utilizes compressed LLMs. Specifically, it assesses how prevalent LLM compression methods affect PEFT performance and subsequently implements strategies for knowledge retention and recovery to counteract the loss of knowledge induced by compression. Lastly, Tied-LoRA (Renduchintala et al., 2023) uses weight tying and selective training to further increase parameter efficiency of LoRA.

**Adapter-based Tuning.** Adapters are bottleneck-like trainable modules integrated into LLMs, which first down-project the input feature vector followed by a non-linear layer and then up-project back to the original size (Houlsby et al., 2019). Adapter-based tuning includes both series adapters and parallel adapters. In series adapters, each LLM layer has two adapter modules added after its attention and feed-forward modules; whereas parallel adapters position two adapter modules alongside the attention and feed-forward modules within each layer of the LLM. In particular, Hu et al. (2023b) propose LLM-Adapters, which integrates series or parallel adapters into LLMs for fine-tuning on different tasks. Karimi Mahabadi et al. (2021) propose Compacter, which unifies adapters, low-rank techniques, and the latest hyper-complex multiplication layers to achieve a balanced trade-off between the amount of trainable parameters and task performance compared to original Adapter method (Houlsby et al., 2019). Furthermore, (IA)$^3$ (Liu et al., 2022a) introduces a technique that scales activations using learned vectors. It outperforms Adapter (Houlsby et al., 2019) and Compacter on few-shot setting with better accuracy and computational efficiency. Following meta-learning principles, Meta-Adapters (Bansal et al., 2022) designs a resource-efficient fine-tuning technique for the few-shot scenario where it incorporates adapter layers that have been meta-learned into a pre-trained model, transforming the fixed pre-trained model into an efficient few-shot learning framework. Meta-Adapters outperforms Adapter (Houlsby et al., 2019) at few-shot fine-tuning with less parameters to fine-tune. AdaMix (Wang et al., 2022c) takes inspiration from sparsely-activated mixture-of-experts (MoE) models (Zuo et al., 2022) and proposes a mixture of adaptation modules to learn multiple views of the given task. Compared to Adapter, it demonstrates better results on both natural language understanding and generation tasks with less learnable parameters. Lastly, OpenDelta (Hu et al., 2023a) is an open-source software library that offers a versatile and plug-and-play framework for implementing a range of adapter-based techniques, and is designed to be compatible with various LLMs architectures.

**Prefix Tuning.** Prefix-Tuning (Li & Liang, 2021) adds a series of trainable vectors, known as prefix tokens, to each layer in an LLM. These prefix tokens are tailored to specific tasks and can be treated as virtual word embeddings. Building on top of Prefix-Tuning, LLaMA-Adapter (Zhang et al., 2024) incorporates a set of trainable adaptation embeddings and attaches them to the word embeddings in the upper layers of the LLMs. A zero-initialized attention scheme with zero gating is also introduced. It dynamically incorporates new guiding signals into LLaMA-1 while retaining its pre-trained knowledge. Different from conventional prefix tuning, HyperTuning (Phang et al., 2023) employs a hyper-model to produce task-specific parameters such as soft prefixes for a downstream model, showing improved performance through initialization from hypermodel-generated parameters for subsequent fine-tuning.

**Prompt Tuning.** Different from prefix tuning, prompt tuning incorporates trainable prompt tokens only at the input layer. These tokens can be inserted either as a prefix or anywhere within the input tokens. Prompt Tuning (Lester et al., 2021) keeps the entire pre-trained model fixed while adding an extra $k$ trainable tokens at the beginning of the input text for each downstream task. It outperforms few-shot prompts and narrows the performance gap compared to full-model fine-tuning. P-Tuning (Liu et al., 2023b) utilizes a small number of parameters as prompts, which are processed by a prompt encoder before being used as input for pre-trained LLMs. Instead of searching for discrete prompts, P-Tuning fine-tunes these prompts through gradient descent and improves performance on a wide range of natural language understanding tasks compared to Prompt Tuning. Liu et al. (2022c) observe that earlier versions of prefix tuning struggle with complex sequence labeling tasks. To address this, they propose P-Tuning v2, which borrows the ideas from prefix tuning by introducing continuous prompts at each layer of the pre-trained model. This modification has proven effective in boosting performance across various parameter sizes for tasks related to natural language understanding. Tam et al. (2023) introduce efficient prompt tuning for text retrieval, updating just 0.1% of parameters and outperforming traditional full-parameter update methods in diverse domains. Sun et al. (2023a) claim that prompt tuning tends to struggle in few-shot learning scenarios, and thus propose MP$^2$ that pre-trains a collection of modular prompts using multitask learning. These prompts are then selectively triggered and assembled by a trainable routing mechanism for specific tasks. As a result, MP$^2$ can quickly adapt to downstream tasks by learning how to merge and reuse pretrained modular prompts. Different from MP$^2$, PPT (Gu et al., 2022b) attributes the performance degradation of prompt tuning in few-shot learning to the poor initialization of soft prompt, and thus proposes to add the soft prompt into the pre-training stage for a better initialization. Lastly, Multitask Prompt Tuning (Wang et al., 2023j) extends Prompt Tuning and harnesses the knowledge of the various tasks through the use of prompt vectors

in a multitask learning settings. Specifically, it initially learns a single, transferable prompt by extracting knowledge from various task-specific source prompts, and then applies multiplicative low-rank updates to this prompt to effectively tailor it for each downstream task. By doing this, Multitask Prompt Tuning is able to attain performance levels that are competitive compared to full-model fine-tuning methods.

### 2.3.2 Memory-Efficient Fine-Tuning

Different from PEFT methods which focus on parameter efficiency, MEFT methods focus on memory savings during the LLMs fine-tuning process. For instance, Dettmers et al. (2023) propose QLoRA which first quantizes the model into a 4-bit NormalFloat data type, and then fine-tunes this quantized model with added low-rank adapter (LoRA) weights (Hu et al., 2022). In doing so, QLoRA reduces memory usage during fine-tuning without performance degradation compared to standard full-model fine-tuning. QA-LoRA (Xu et al., 2024b) improves QLoRA by introducing group-wise operators that improve quantization flexibility (each group is quantized separately) while reducing adaptation parameters (each group utilizes shared adaptation parameters). Similarly, LoftQ (Li et al., 2024d) combines model quantization with singular value decomposition (SVD) to approximate the original high-precision pre-trained weights. As a result, it offers a favorable initialization point for subsequent LoRA fine-tuning, leading to enhancements over QLoRA on both natural language understanding and generation tasks. PEQA (Kim et al., 2023a) introduces a two-stage approach to quantization-aware fine-tuning. In the first stage, the parameter matrix for each fully connected layer is quantized into a matrix of low-bit integers along with a scalar vector. In the second stage, the low-bit matrix remains unchanged, while fine-tuning is focused solely on the scalar vector for each specific downstream task. Employing this two-stage approach, PEQA not only minimizes memory usage during fine-tuning but also speeds up inference time by maintaining weights in a low-bit quantized form, showing better perplexity than GPTQ (Frantar et al., 2023) with LoRA. Different from above-mentioned MEFT methods that combine LoRA with quantization to reduce fine-tuning memory footprints, as another line of research, some studies propose MEFT methods based on gradient optimization. Specifically, Simoulin et al. (2023) propose Selective Fine-Tuning which minimizes memory usage by specifically preserving a subset of intermediate activations from the forward pass for which the calculated gradients are nonzero. Notably, this approach delivers performance equivalent to full-model fine-tuning while using just up to one-third of the GPU memory required otherwise. Lv et al. (2023) introduce LOMO, which minimizes memory consumption during fine-tuning by combining gradient calculation and parameter updating into a single step. As such, LOMO eliminates all components of the optimizer state, lowering the memory requirements for gradient tensors to $O(1)$. Lastly, MeZO (Malladi et al., 2023) improves the zeroth-order method (Spall, 1992) for gradient estimation using only two forward passes. This enables efficient fine-tuning of LLMs with memory requirements similar to inference and supports both full-parameter and PEFT methods like LoRA (Hu et al., 2022) and prefix tuning (Li & Liang, 2021), enabling MeZO to train a 30-billion parameter model on a single A100 80GB GPU.

### 2.4 Efficient Inference

Efficient inference techniques focus on reducing the costs of the LLMs inference process. As summarized in Figure 10, efficient inference techniques can be grouped into techniques at algorithm level and system level.

**Algorithm-Level Inference Efficiency Optimization.** Techniques that enhance LLM inference efficiency at the algorithm level include speculative decoding and KV-cache optimization.

- ***Speculative Decoding.*** Speculative decoding (i.e., speculative sampling) (Leviathan et al., 2023) is a decoding strategy for autoregressive language models that speeds up the sampling process by computing tokens using a smaller draft model in parallel to create speculative prefixes for the large target model. Chen et al. (2023a) focus on the distributed serving setting for LLMs and propose to run a faster autoregressive model $K$ times and then evaluate the preliminary output with the large target model. A tailored rejection sampling strategy is employed to approve a selection of the draft tokens in a left-to-right order, thereby recapturing the distribution of the large target model during the procedure. Staged Speculative (Spector & Re, 2023) transforms the speculative batch into a tree structure representing potential token sequences. This restructuring expedites the

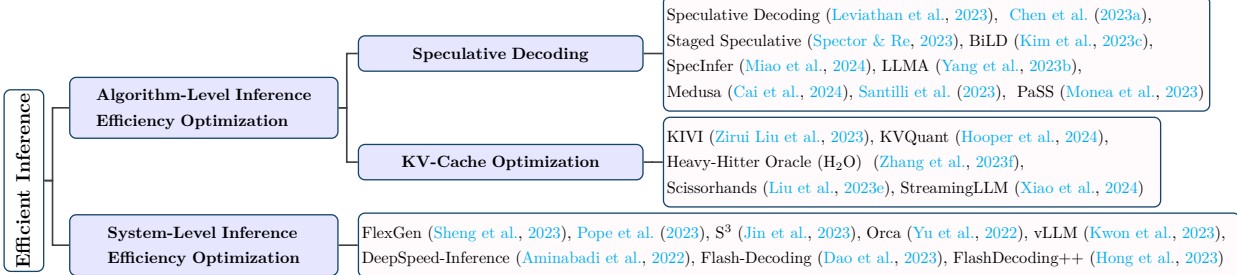

Figure 10: Summary of efficient inference techniques for LLMs.

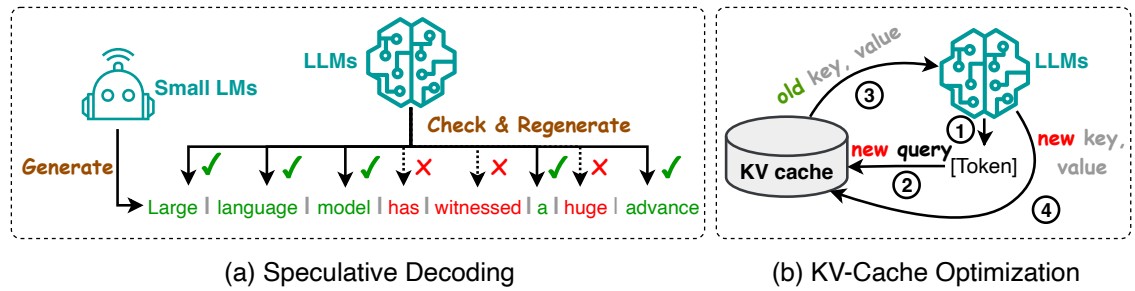

(a) Speculative Decoding     (b) KV-Cache Optimization

Figure 11: Illustrations of algorithm-level efficiency optimization techniques for LLM inference.

generation of larger and improved speculative batches. It also introduces an additional phase for speculative decoding of the initial model, thereby enhancing overall performance, showing 1.36x over standard speculative decoding. BiLD (Kim et al., 2023c) optimizes speculative decoding through two innovative techniques: the fallback policy that permits the smaller draft model to waive control to the larger target model when it lacks sufficient confidence; and the rollback policy that enables the target model to revisit and rectify any inaccurate predictions made by the smaller draft model. SpecInfer (Miao et al., 2024) extends speculative decoding and speeds up inference by employing speculative inference techniques and token tree validation. Its core idea involves merging a range of small speculative models that have been fine-tuned collectively to collaboratively forecast the output of the large target model, which is then used to validate all the predictions. Different from speculative decoding that needs to introduce an additional efficient drafter model to generate a draft for checking, LLMA (Yang et al., 2023b) chooses a text segment from a closely related reference and duplicates its tokens into the decoder. It then concurrently assesses the suitability of these tokens as the decoding output within a single decoding step. This approach results in a speed increase of more than two times while maintaining the same generated results as traditional greedy decoding. Similarly, instead of using a separate draft model to sequentially generate candidate output, Medusa (Cai et al., 2024) proposes to freeze the LLM backbone, fine-tune additional heads, and use a tree-based attention mechanism to process predictions in parallel to speed up the decoding process. Lastly, Santilli et al. (2023) propose parallel decoding including the Jacobi and Gauss-Seidel fixed-point iteration methods for speculative decoding. Among these methods, Jacobi decoding was extended into Lookahead decoding (Fu et al., 2023c) to further enhance the efficiency.

- **KV-Cache Optimization.** During inference, LLMs need to store the Key-Value (KV) pairs of the past tokens into the cache for future token generation. The size of KV cache needed enlarges massively with the increase of generated token length, resulting in considerable memory consumption and long inference latency. Therefore, reducing the size of KV cache is key to enhancing inference efficiency. Existing KV-cache optimization techniques can in general be grouped into two categories. The first category is to compress the KV cache. For example, Zirui Liu et al. (2023) propose KIVI, a tuning-free 2bit KV cache quantization algorithm which quantizes the key cache per-channel and the value cache per-token, achieving 2.6x less peak memory usage during inference. Similarly, Hooper

et al. (2024) conduct an empirical study on the impact of per-channel quantization and other types of quantization such as quantization before rotary positional embedding. Based on their findings, they propose KVQuant which combines these quantization methods to quantize the KV cache of LLaMA to 3-bit. The second category of KV-Cache optimization techniques is to evict some KVs from the cache. For instance, Zhang et al. (2023f) propose Heavy-Hitter Oracle ($H_2O$), a KV cache eviction strategy that formulates the KV cache eviction as a dynamic submodular problem and dynamically retains a balance between recent and performance-critical tokens, improving the throughput for LLMs inference. Similarly, Liu et al. (2023e) propose a hypothesis named the persistence of importance, suggesting that only tokens that were crucial at an earlier phase will have a significant impact on subsequent stages. Based on this hypothesis, they design Scissorhands which significantly reduces the KV cache without compromising model quality. Lastly, StreamingLLM (Xiao et al., 2024) incorporates window attention, where only the most recent KVs are cached into a fixed-size sliding window, into their algorithm design. Through evicting the outdated KVs, StreamingLLM ensures constant memory usage and decoding speed after the cache is initially filled.

**System-Level Inference Efficiency Optimization.** The efficiency of LLM inference can also be optimized at the system level under a specific hardware architecture. For example, FlexGen (Sheng et al., 2023) is a high-throughput inference engine that enables the execution of LLMs on GPUs with limited memory. It uses a linear programming-based search approach to coordinate various hardware, combining the memory and computation from GPU, CPU, and disk. Furthermore, FlexGen quantizes the weights and attention cache to 4 bits, which increases the inference speed of OPT-175B (Zhang et al., 2022a) on a single 16GB GPU. Pope et al. (2023) develop a simple analytical framework to partition a model in order to scale Transformer inference based on the application requirements. By combining it with scheduling and memory optimizations, they are able to achieve better efficiency on PaLM (Chowdhery et al., 2022) in comparison to FasterTransformer (NVIDIA, 2023a). Orca (Yu et al., 2022) employs iteration-level scheduling to serve batched sequences with variable output sequence length. When a sequence in a batch is completed, it is returned to the user so that a new sequence can be served immediately. As a result, Orca improves GPU utilization compared to static batching, showing $36.9\times$ throughput improvement under the same level of latency compared to FasterTransformer. $S^3$ (Jin et al., 2023) creates a system that is aware of the output sequence beforehand. It can anticipate the length of the sequence and arrange generation requests accordingly, optimizing the utilization of device resources and increasing the rate of production, showing higher throughput than Orca with the same number of GPUs. However, both Orca and $S^3$ lead to memory fragmentation due to their inaccurate memory provisioning for each request. vLLM (Kwon et al., 2023) addresses the memory efficiency problem with PagedAttention, which enables the storage of continuous keys and values in non-contiguous memory space. Specifically, PagedAttention divides the KV cache of each sequence into blocks, each containing the keys and values for a fixed number of tokens. During attention computation, PagedAttention kernel manages these blocks efficiently by maintaining a block table to reduce memory fragmentation. Specifically, the contiguous logical blocks of a sequence are mapped to non-contiguous physical blocks via the table and the table automatically allocates a new physical block for every newly generated token. This reduces the amount of memory wasted when generating new tokens, thus improving its efficiency, showing that PagedAttention improves the throughput of popular LLMs by 2-4$\times$ with the same level of latency compared to FasterTransformer (NVIDIA, 2023a) and Orca (Yu et al., 2022). On the other hand, infinitely optimizing server-side aggregated metrics does not necessarily lead to good user experience or Quality of Experience (QoE) especially under high server load. Andes (Liu et al., 2024b) first defines QoE for the LLM-based text streaming services and proposes a QoE-aware serving systems to optimize QoE by prioritizing requests based on their resource demand and service acquired. DeepSpeed-Inference (Aminabadi et al., 2022) is a multi-GPU inference approach designed to enhance the efficiency of both dense and sparse Transformer models when they are contained within the collective GPU memory. Furthermore, it provides a mixed inference technique that utilizes CPU and NVMe memory, in addition to GPU memory and computation, enabling high-throughput inference even for models that are too large to fit in the combined GPU memory. This approach demonstrates lower latency than FasterTransformer under the same throughput. Flash-Decoding (Dao et al., 2023) boosts the speed of long-context inference by breaking down keys/values into smaller pieces, computing attention on these pieces in parallel, and then combining them to generate the final output. It outperforms FasterTransformer and FlashAttention (Dao

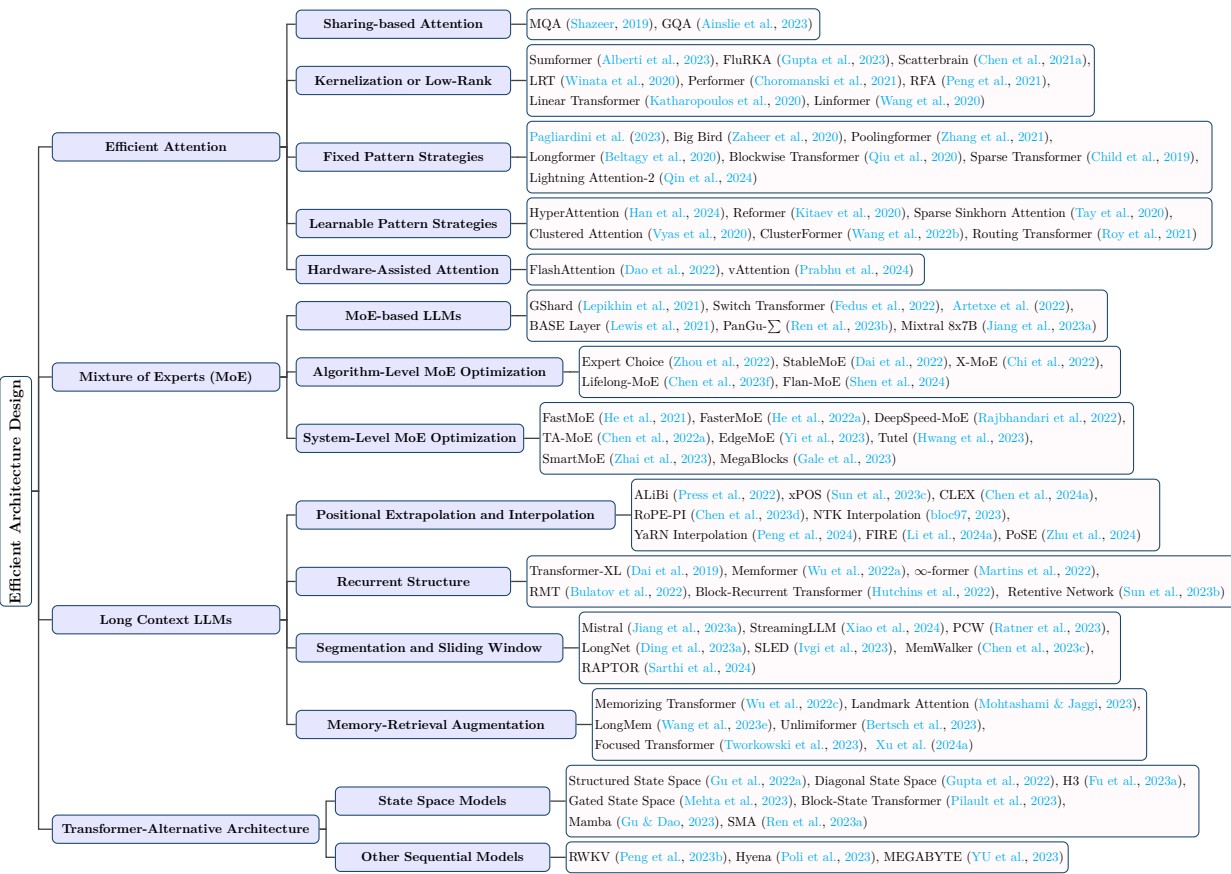

Figure 12: Summary of efficient architecture designs for LLMs.

et al., 2022) in decoding speed for very large sequences. FlashDecoding++ (Hong et al., 2023) supports mainstream language models and hardware backends through asynchronous softmax, double buffering for flat GEMM optimization, and heuristic dataflow, resulting in up to 4.86x and 2.18x acceleration on Nvidia and AMD GPUs respectively compared to HuggingFace implementations, showing higher speedup compared to Flash-Decoding under the same throughput.

## 2.5 Efficient Architecture Design

Efficient architecture design for LLMs refers to the strategic optimization of model architecture and computational processes to enhance performance and scalability while minimizing resource consumption. Figure 12 provides a summary of existing efforts on designing efficient architectures for LLMs.

### 2.5.1 Efficient Attention

The quadratic time and space complexity of attention modules considerably slows down the pre-training, inference, and fine-tuning of LLMs (Duman Keles et al., 2023). Many techniques have been proposed to make attention lightweight for more efficient execution. These techniques can be generally categorized as sharing-based attention, kernelization or low-rank, fixed pattern strategies, learnable pattern strategies, and hardware-assisted attention.

**Sharing-based Attention.** Sharing-based attention accelerates attention computation during inference through KV heads sharing. For example, LLaMA-2 (Touvron et al., 2023b) optimizes the autoregressive decoding process by using multi-query attention (MQA) (Shazeer, 2019) and grouped-query attention (GQA) (Ainslie et al., 2023). In traditional multi-head attention (MHA), each head has distinct linear

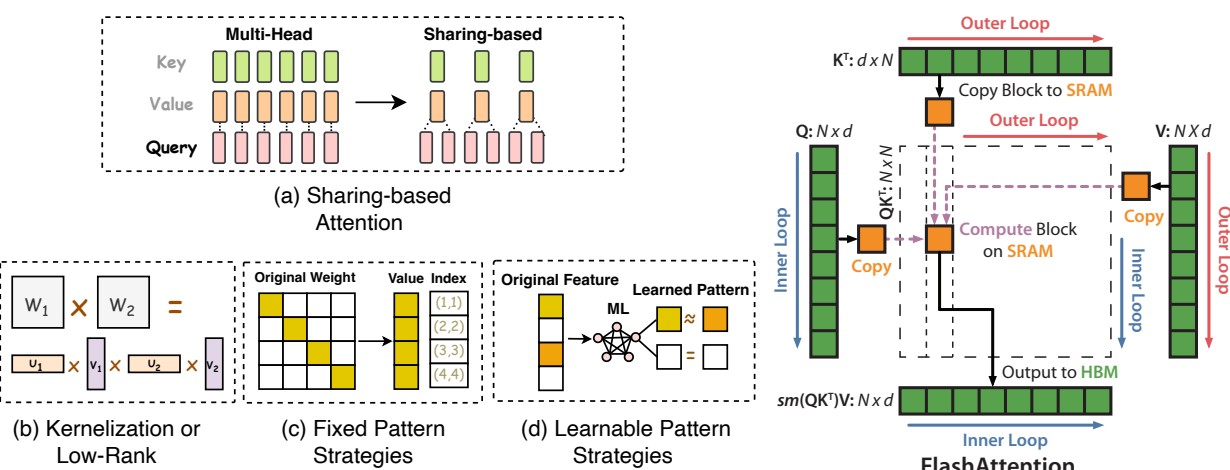

Figure 13: Illustrations of attention optimizations.

Figure 14: Design of FlashAttention (Dao et al., 2022).

transformations for input matrix queries (Q), keys (K) and values (V), allowing for diverse representations and attention mechanisms across different subspaces. In MQA, all heads share a single set of key and value weights across all query heads. Thus, MQA speeds up the inference but could compromise output quality. To address this drawback, GQA interpolates MQA and MHA by employing one key and value heads for each group of query heads to enhance inference quality.

**Kernelization or Low-Rank.** Kernelization or low-rank techniques adopted by models such as Sumformer (Alberti et al., 2023), FluRKA (Gupta et al., 2023), Scatterbrain (Chen et al., 2021a), Low-Rank Transformer (LRT) (Winata et al., 2020), Performer (Choromanski et al., 2021), Random Feature Attention (RFA) (Peng et al., 2021), Linear Transformer (Katharopoulos et al., 2020), and Linformer (Wang et al., 2020) enhance the efficiency by utilizing low-rank representations of the self-attention matrix or by adopting attention kernelization techniques. Specifically, low-rank methods focus on compacting the dimensions of attention keys and values. For example, Linformer (Wang et al., 2020) proposes to segment scaled dot-product attention into smaller units via linear projection. Kernelization, a variant of low-rank techniques, focuses on approximating the attention matrix (Choromanski et al., 2020). For example, Performer (Choromanski et al., 2021) condenses softmax attention-kernels using positive orthogonal random features, outperforming Reformer and Linformer on long protein sequence benchmark. Sumformer (Alberti et al., 2023) approximates the equivariant sequence-to-sequence function, offering a universal solution for Linformer and Performer.

**Fixed Pattern Strategies.** Fixed pattern strategies adopted by models such as (Pagliardini et al., 2023), Big Bird (Zaheer et al., 2020), Poolingformer (Zhang et al., 2021), Longformer (Beltagy et al., 2020), Blockwise Transformer (Qiu et al., 2020), and Sparse Transformer (Child et al., 2019) improve efficiency by sparsifying the attention matrix. This is achieved by confining the attention scope to predetermined patterns, such as local windows or fixed-stride block patterns. For instance, the attention mechanism adopted by Longformer (Beltagy et al., 2020), designed as an alternative to conventional self-attention, merges local windowed attention with globally oriented attention tailored to specific tasks. Pagliardini et al. (2023) expand FlashAttention (Dao et al., 2022) to support a broad spectrum of attention sparsity patterns, including key-query dropping and hashing-based attention techniques, achieving a multi-fold runtime speedup on top of FlashAttention on long text benchmark.

**Learnable Pattern Strategies.** Different from fixed pattern strategies, learnable pattern strategies adopted by models such as HyperAttention (Han et al., 2024), Reformer (Kitaev et al., 2020), Sparse Sinkhorn Attention (Tay et al., 2020), Clustered Attention (Vyas et al., 2020), ClusterFormer (Wang et al., 2022b), and Routing Transformer (Roy et al., 2021) improve efficiency by learning token relevance and subsequently grouping tokens into buckets or clusters. As an example, HyperAttention (Han et al., 2024) proposes a parameterization for spectral approximation and employs two key metrics: the maximal column norm in the

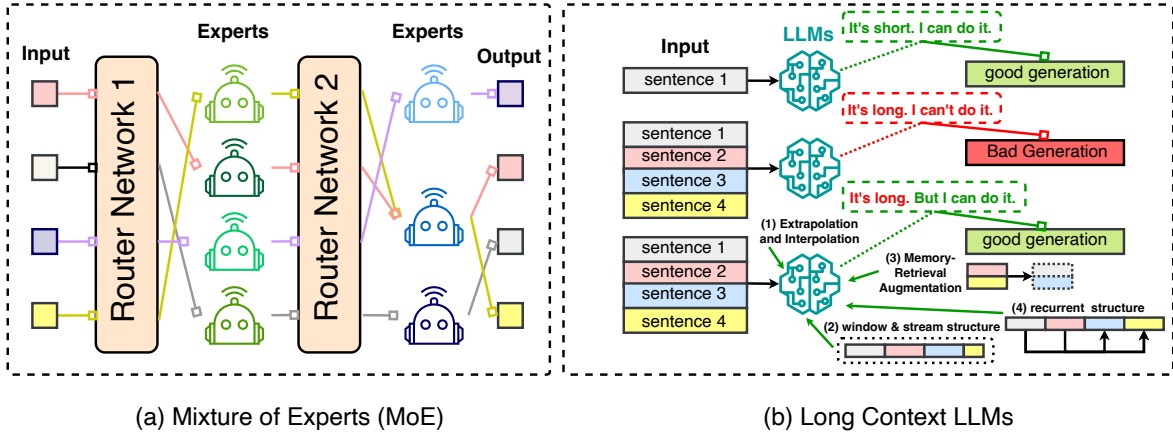

(a) Mixture of Experts (MoE)

(b) Long Context LLMs

Figure 15: Illustrations of Mixture of Experts (MoE) and long context LLMs.

normalized attention matrix and the row norm ratio in the unnormalized matrix after large entry removal. It also utilizes the learnable sort locality-sensitive hashing (sortLSH) technique and fast matrix multiplication via row norm sampling. The experiment results show that HyperAttention enhances both inference and training speeds for LLMs with only minimal performance degradation, giving significant speed improvements compared to FlashAttention (Dao et al., 2022) on long contexts.

**Hardware-Assisted Attention.** Hardware-assisted attention focuses on developing hardware-specific techniques to enhance attention efficiency. For example, FlashAttention (Dao et al., 2022) reduces the number of memory access between GPU high-bandwidth memory (HBM) and GPU on-chip SRAM when calculating the attention module in LLMs. Instead of transmitting the values and results between HBM and SRAM multiple times as is done in the standard attention mechanism, FlashAttention combines all the attention operations into one kernel and tiles the weight matrices into smaller blocks to better fit the small SRAM as shown in Figure 14. As a result, only one communication is required to process each attention block, which significantly enhances the efficiency for processing the entire attention block. vAttention (Prabhu et al., 2024) is proposed to store KV cache in contiguous virtual memory without committing physical memory ahead-of-time. It avoids the software complexity to store KV cache by leveraging CUDA support of low-level virtual memory APIs.

### 2.5.2 Mixture of Experts (MoE)

Mixture of Experts (MoE) represents a sparse methodology utilized prominently in large-scale models like LLMs. It operates on the principle of segmenting a designated task into several sub-tasks, and then developing numerous smaller, specialized models, dubbed *experts*, with each honing in on a distinct sub-task. Subsequently, these experts collaborate to deliver a consolidated output. For pre-traning or fine-tuning, MoE requires developers to manage a huge number of parameters efficiently, enhancing the model's capacity and potentially its performance while keeping the computational and memory requirements relatively manageable. For inference, MoE decreases the inference time by not engaging all experts simultaneously, but rather activating only a select few. Additionally, MoE is capable of minimizing communication between devices in model-distributed scenarios by allocating each expert to an individual accelerator; communication is only necessary between the accelerators that host the router and the relevant expert model (Kaddour et al., 2023).

**MoE-based LLMs.** Several MoE-based LLMs have been proposed. For example, GShard (Lepikhin et al., 2021) is a MoE-based LLM that offers a refined method to articulate a variety of parallel computation frameworks with minor modifications to the existing model code. It also amplifies a multilingual neural machine translation Transformer model with Sparsely-Gated MoE beyond 600 billion parameters through automatic sharding. Switch Transformer (Fedus et al., 2022) brings forth a switch routing algorithm and crafts intuitively enhanced models, lowering communication and computational expenditures. It encompasses up to one trillion parameters, dividing tasks among up to 2,048 experts, thereby illustrating the scalability

and efficacy of the MoE framework. Artetxe et al. (2022) scale sparse language models to 1.1T parameters, discerning superior performance up to this scale in language modeling, zero-shot and few-shot learning in comparison to dense models. This suggests that sparse MoE models are a computationally efficient substitute for traditionally employed dense architectures. Its largest MoE model outperforms its dense counterpart where the latter requires twice as much computation. BASE Layer (Lewis et al., 2021) defines token-to-expert allocation as a linear assignment problem, allowing an optimal assignment where each expert acquires an equal number of tokens, achieving lower validation perplexity during training relative to Switch Transformer. PanGu-Σ (Ren et al., 2023b) is a MoE-based LLM with 1.085T parameters, transitioned from the dense Transformer model to a sparse one with Random Routed Experts (RRE), and effectively trains the model over 329B tokens utilizing Expert Computation and Storage Separation (ECSS). It outperforms dense model like ERNIE 3.0 Titan Wang et al. (2021) on zero-shot test of Chinese downstream task. Lastly, Mixtral 8x7B (Jiang et al., 2023a) is a MoE with 46.7B total parameters. By leveraging the advantage of MoE architecture, Mixtral 8x7B outperforms LLaMA-2 70B on most benchmarks such as MMLU, MBPP, and GSM-8K with 6x faster inference by only using 12.9B parameters of the model per token for inference.

**Algorithm-Level MoE Optimization.** The efficiency of MoE-based LLMs can be improved at the algorithm level. Expert Choice (Zhou et al., 2022) allows experts to pick the top-k tokens instead of having tokens choose the top-k experts, implying that each token can be directed to a variable number of experts while each expert maintains a fixed bucket size. This method demonstrates higher performance in the GLUE and SuperGLUE benchmarks, and outperforms T5 dense model in seven out of the 11 tasks. StableMoE (Dai et al., 2022) identifies the issue of altering target experts for identical input during training and addresses this by creating two training phases. Initially, it cultivates a balanced routing strategy, which is then distilled into a decoupled lightweight router. In the following phase, this distilled router is used for a fixed token-to-expert assignment, ensuring a stable routing strategy. StableMoE shows better results than Switch Transformer and BASE Layer with lower validation perplexity on language modeling. X-MoE (Chi et al., 2022) notes that earlier routing mechanisms foster token clustering around expert centroids, indicating a tendency toward representation collapse. It proposes to estimate the routing scores between tokens and experts on a low-dimensional hyper-sphere, showing improvements over Switch Transformer on multilingual multi-task benchmark. Lifelong-MoE (Chen et al., 2023f) observes that MoE increases the capacity of the model to adapt to different corpus distributions in online data streams without extra computational cost, simply by incorporating additional expert layers and suitable expert regularization. This facilitates continuous pre-training of a MoE-based LLM on sequential data distributions without losing previous knowledge. It outperforms other MoE models such as GShard on natural language generation and understanding tasks. Lastly, Shen et al. (2024) observe that compared to dense models, MoE gains more from instruction tuning. Based on this observation, they propose Flan-MoE, which combines MoE and instruction tuning to enlarge language models without increasing demands in memory and compute resources while showing better zero-shot and few-shot performance compared to FLAN-T5 dense model.

**System-Level MoE Optimization.** The efficiency of MoE-based LLMs can also be improved at the system level. For example, FastMoE (He et al., 2021) is a distributed MoE training system built on PyTorch, compatible with common accelerators. This system offers a hierarchical interface that allows both flexible model design and easy adaptation to various applications, such as Transformer-XL and Megatron-LM. FasterMoE (He et al., 2022a) tries to address the challenges of dynamic load imbalance, inefficient synchronous execution mode, and congested all-to-all communication during MoE training. It first introduces a performance model that predicts latency and analyzes end-to-end performance through a roofline-like methodology. Utilizing this model, it presents a dynamic shadowing technique for load balancing, a concurrent fine-grained schedule for operations, and a strategy to alleviate network congestion by adjusting expert selection for model training. It outperforms FastMoE and achieves $1.37\times$ - $17.87\times$ speedup compared to methods including ZeRO, GShard, and BASE Layer. Lina (Li et al., 2023a) also improves training efficiency and inference time by optimizing communication and load balancing in distributed MoE. Lina first prioritizes all-to-all over allreduce using tensor partitioning and pipelining to improve its bandwidth in training, and then dynamically balances the workload with token-level expert selection pattern in inference. DeepSpeed-MoE (Rajbhandari et al., 2022) has designed a Pyramid-Residual MoE (PR-MoE) architecture to enhance both the training and the inference efficiency of the MoE model parameter. PR-MoE is a dense-MoE hybrid that employs residual connections to optimally utilize experts, managing to reduce the parameter size by

up to 3x without sacrificing quality or compute requirements. It serves massive MoE models with up to 4.5x faster and 9x cheaper inference compared to quality-equivalent dense models. DeepSpeed-MoE also proposes a highly optimized MoE inference system which enables efficient scaling of inference workloads on hundreds of GPUs, providing up to 7.3x reduction in inference latency and cost when compared with existing MoE inference solutions. TA-MoE (Chen et al., 2022a) highlights that current MoE dispatch patterns do not fully leverage the underlying heterogeneous network environment and thus introduces a topology-aware routing strategy for large-scale MoE training that dynamically modifies the MoE dispatch pattern based on the network topology, making it outperform FastMoE, FasterMoE, and DeepSpeed-MoE. EdgeMoE (Yi et al., 2023) presents an on-device inference engine tailored for MoE-based LLMs. It optimizes memory and computation for inference by distributing the model across different storage levels. Specifically, non-expert model weights are stored directly on the edge device, while expert weights are kept externally and only loaded into the device's memory when necessary. Tutel (Hwang et al., 2023) proposes adaptive parallelism and pipelining features to adapt to the dynamic workload. It employs a consistent layout for MoE parameters and input data, supporting switchable parallelism and dynamic pipelining without any mathematical inconsistencies or tensor migration costs, thus enabling free run-time optimization, achieving up to $5.75\times$ speedup for a single MoE layer. SmartMoE (Zhai et al., 2023) focuses on the automatic parallelization for MoE distributed training. In the offline stage, SmartMoE constructs a search space of hybrid parallelism strategies. In the online stage, it incorporates light-weight algorithms to identify the optimal parallel strategy. It achieves up to $1.88\times$ speedup in end-to-end training over FasterMoE on a distributed training setting. Lastly, MegaBlocks (Gale et al., 2023) transforms MoE-oriented computation with block-sparse operations and creates block-sparse GPU kernels to optimize MoE computation on hardware. This leads to training time up to 40% shorter compared to Tutel and 2.4x shorter than dense models trained with Megatron-LM.

### 2.5.3 Long Context LLMs

In many real-world applications, such as multi-turn conversations and meeting summarization, existing LLMs are often required to comprehend or generate context sequences that are much longer than what they have been pre-trained with and may result in a degradation in accuracy due to the poor memorization for the long context. One direct way to address this issue is to fine-tune LLMs with similar long-sequence data, which, however, is time consuming and computation-intensive. To fill this gap, new methods have been developed to enable LLMs to adapt to longer context lengths in a more efficient way. These methods can be in general grouped into four categories: extrapolation and interpolation, recurrent structure, segmentation and sliding window, and memory-retrieval augmentation.

**Positional Extrapolation and Interpolation.** Standard positional encoding methods such as absolute positional embeddings (APE) (Vaswani et al., 2017), learned positional embeddings (LPE) (Wang et al., 2022a), relative positional embeddings (RPE) (Shaw et al., 2018), and rotary position embeddings (RoPE) (Su et al., 2023b) have advanced the integration of positional information in LLMs. For example, LPE has been used by GPT-3 and OPT, RPE was used by Gopher (Rae et al., 2022) and Chinchilla (Hoffmann et al., 2022), whereas RoPE was used by LLaMA-1 and GLM-130B. However, it is still challenging to train LLMs on sequences with a limited maximum length while still ensuring them to generalize well on significantly longer sequences during inference. Given that, techniques based on positional extrapolation (Press et al., 2022; Sun et al., 2023c; Chen et al., 2024a) and positional interpolation (Chen et al., 2023d; Peng et al., 2024; Li et al., 2024a) have been proposed.

Positional extrapolation strategies extend the encoding of positional information beyond what the model has explicitly learned during training. For example, ALiBi (Press et al., 2022) applies attention with linear biases to attain extrapolation for sequences that exceed the maximum length seen during training. By applying negatively biased attention scores with a linearly diminishing penalty based on the distance between the pertinent key and query, it facilitates efficient length extrapolation. Different from ALiBi, xPOS (Sun et al., 2023c) characterizes attention resolution as a marker for extrapolation and utilizes a relative position embedding to enhance attention resolution, thereby improving length extrapolation. However, these techniques have not been implemented in some of the recent LLMs such as GPT-4, LLaMA, and LLaMA-2. CLEX (Chen et al., 2024a) proposes to generalize position embedding scaling with ordinary differential

equations to model continuous dynamics over length scaling factors. In doing so, CLEX gets rid of the limitations of existing positional extrapolation scaling methods to enable long-sequence generation.

Positional interpolation strategies, on the other hand, reduce the scale of input position indices and extend the context window sizes, allowing LLMs to maintain their performance over longer text sequences. For example, Chen et al. (2023d) observe that extending beyond the trained context length could impair the self-attention mechanism. They propose RoPE-PI to reduce the position indices through linear interpolation, aligning the maximum position index with the prior context window limit encountered during pre-training. NTK interpolation (bloc97, 2023) modifies the base of the RoPE, effectively changing the rotational velocity of each RoPE dimension. YaRN interpolation (Peng et al., 2024) uses a ramp function to blend linear and NTK interpolation in varying proportions across dimensions and incorporates a temperature factor to counteract distribution shifts in the attention matrix caused by long inputs. Experimental results on long-text modeling shows that YaRN outperforms existing RoPE interpolation methods including RoPE-PI and NTK. FIRE (Li et al., 2024a) proposes a functional relative position encoding using learnable mapping of input positions to biases and progressive interpolation, ensuring bounded input for encoding functions across all sequence lengths to enable length generalization. It demonstrates competitive results compared to ALiBi, RoPE, and RoPE-PI on long text benchmarks. Lastly, PoSE (Zhu et al., 2024) proposes positional skip-wise training that simulates long inputs using a fixed context window and designs distinct skipping bias terms to manipulate the position indices of each chunk. This strategy reduces memory and time consumption compared to full-length fine-tuning.

**Recurrent Structure.** LLMs' ability to manage long sequences can also be enhanced through recurrence structure. For example, Transformer-XL (Dai et al., 2019) presents a segment-level recurrence mechanism and utilizes enhanced relative positional encoding to capture long-term dependencies and address the long-context fragmentation issue. Memformer (Wu et al., 2022a) leverages an external dynamic memory for encoding and retrieving past information, achieving linear time and constant memory space complexity for long sequences. It also proposes Memory Replay Back-Propagation (MRBP) to facilitate long-range back-propagation through time with significantly lower memory requirements, achieving better results than Transformer-XL on language modeling and image generation benchmarks. $\infty$-former (Martins et al., 2022) presents a Transformer model augmented with unbounded long-term memory (LTM). It employs a continuous space attention framework to balance the quantity of information units accommodated in memory against the granularity of their representations, showing better results than Transformer-XL on long text sorting and modeling. Recurrent Memory Transformer (RMT) (Bulatov et al., 2022) uses a recurrence mechanism to retain information from the past segment level by incorporating special memory tokens into the input or output sequence, and demonstrates superior performance compared to Transformer-XL in long context modeling. Block-Recurrent Transformer (BRT) (Hutchins et al., 2022) utilizes self-attention and cross-attention to execute a recurrent function across a broad set of state vectors and tokens so as to model long sequences through parallel computation. Lastly, Retentive Network (Sun et al., 2023b) introduces a multi-scale retention mechanism as an alternative to multi-head attention. By leveraging parallel and chunk-wise recurrent representations, it enables effective scaling, achieves training parallelization and constant inference cost, and offers linear long-sequence memory complexity compared to other Transformer models.

**Segmentation and Sliding Window.** Segmentation and sliding window techniques tackle the issue of long-context processing by dividing the input data into smaller segments, or applying a moving window to slide through the long sequence. For instance, Mistral (Jiang et al., 2023a) uses sliding window attention to handle sequences of arbitrary length with a reduced inference cost. StreamingLLM (Xiao et al., 2024) identifies an attention sink phenomenon, noting that retaining the Key-Value of initial tokens significantly restores the performance of window attention. Based on this observation, it suggests an efficient framework via merging window context and the first token, allowing LLMs trained with a finite length attention window, but have the ability to generalize to infinite sequence lengths without any fine-tuning. Parallel Context Windows (PCW) (Ratner et al., 2023) segments a long context into chunks, limiting the attention mechanism to function only within each window, and then re-deploys the positional embeddings across these windows. LongNet (Ding et al., 2023a) proposes dilated attention, which exponentially expands the attentive field as the distance increases, enabling the handling of sequence lengths of more than one billion tokens. SLED (Ivgi et al., 2023) handles long sequences by partitioning the long text into small chunks and leveraging

pretrained short-text language models for encoding and decoding. Different from the methods mentioned above, MemWalker (Chen et al., 2023c) transforms lengthy texts into segmented summaries within a tree structure, leveraging LLMs as an interactive entity for guided reading through iterative prompts. It outperforms traditional methods based on extended context, recurrence, and retrieval. Similar to MemWalker, RAPTOR (Sarthi et al., 2024) utilizes text chunks to construct a recursive tree to enable information integration from extensive documents across various abstraction levels during inference, achieving superior performance in multi-step reasoning.

**Memory-Retrieval Augmentation.** Lastly, several studies tackle the inference of extremely long text by employing memory-retrieval augmentation strategies. A notable example is the Memorizing Transformer (Wu et al., 2022c), which extends the attention context size by utilizing k-nearest-neighbor (KNN) lookup to fetch previously similar context embeddings. Additionally, Landmark Attention (Mohtashami & Jaggi, 2023) employs a landmark token to represent each block of input and trains the attention mechanism to utilize it for choosing relevant blocks. This allows the direct retrieval of blocks through the attention mechanism while maintaining the random access flexibility of the previous context, demonstrating comparable perplexity as Transformer-XL while reducing FLOPs for long-context modeling. LongMem (Wang et al., 2023e) proposes a decoupled network architecture with the original backbone LLM as a memory encoder and an adaptive residual side network as a memory retriever and reader, efficiently caching and updating long-term past contexts to prevent knowledge staleness. It outperforms Memorizing Transformer on long text modeling and natural language understanding tasks. Unlimiformer (Bertsch et al., 2023) enhances the KNN-augmented Transformer by outputting attention dot-product scores as KNN distances to enable indexing of virtually unlimited input sequences. Experimental results show that Unlimiformer outperforms Memorizing Transformer on long document summarization benchmarks. Tworkowski et al. (2023) observe that the ratio of relevant keys to irrelevant ones diminishes as context length increases. Based on this observation, they propose Focused Transformer (FoT), a contrastive learning-based technique to refine the structure of the Key-Value space. Unlike Memorizing Transformer and Transformer-XL, FoT does not require training on long sequences and shows better performance on both long context and short context tasks. Lastly, Xu et al. (2024a) discover that an LLM with a 4K context window, when augmented with simple retrieval during generation, can match the performance of a fine-tuned LLM with a 16K context window using positional interpolation (Chen et al., 2023d) on long context tasks while requiring significantly less computation.

### 2.5.4 Transformer-Alternate Architectures

While Transformer-based architectures are now at the forefront of LLMs, some studies propose new architectures to supplant Transformer-based architectures.

**State Space Models**. A promising approach that aims to substitute the attention mechanism is state space models (SSMs). SSM is formulated as $x'(t) = Ax(t) + Bu(t)$, $y(t) = Cx(t) + Du(t)$, which maps a single-dimension input signal $u(t)$ to an N-dimension latent state $x(t)$ before projecting to a single-dimension output signal $y(t)$, where $A$, $B$, $C$, $D$ are parameters learned by gradient descent (Gu et al., 2022a). Compared to attention that has quadratic complexity, SSMs provide near-linear computational complexity related to the length of the sequence. Given such advantage, a series of techniques have been proposed to improve SSMs. For example, the Structured State Space (S4) sequence model (Gu et al., 2022a) refines SSMs by conditioning matrix $A$ with a low-rank correction. This enables stable diagonalization and simplifies the SSM to the well-studied computation of a Cauchy kernel. Diagonal State Space (DSS) (Gupta et al., 2022) improves SSMs by proposing fully diagonal parameterization of state spaces instead of a diagonal plus low rank structure. To bridge the gap between SSMs and attention while adapting to modern hardware, H3 (Hungry Hungry Hippo) (Fu et al., 2023a) stacks two SSMs to interact with their output and input projection, allowing it to log tokens and facilitate sequence-wide comparisons. Mehta et al. (2023) introduce a more efficient layer called Gated State Space (GSS), which has been empirically shown to be 2 to 3 times faster than DSS while maintaining the perplexity on multiple language modeling benchmarks. Block-State Transformer (BST) (Pilault et al., 2023) designs a hybrid layer that combines an SSM sublayers for extended range contextualization with a Block Transformer sublayer for short-term sequence representation. Gu & Dao (2023) propose Mamba to enhance SSMs by designing a selection mechanism to eliminate irrelevant data and develop a hardware-aware parallel algorithm for recurrent operation, achieving 5x throughput than

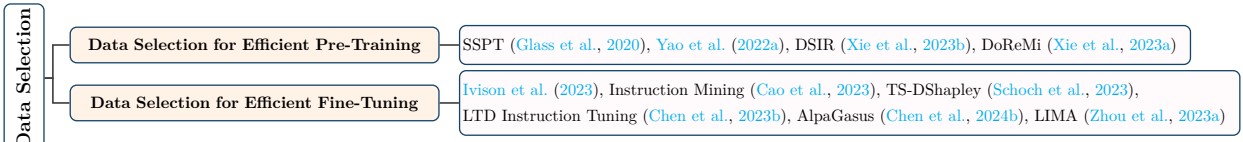

Figure 16: Summary of data selection techniques for LLMs.

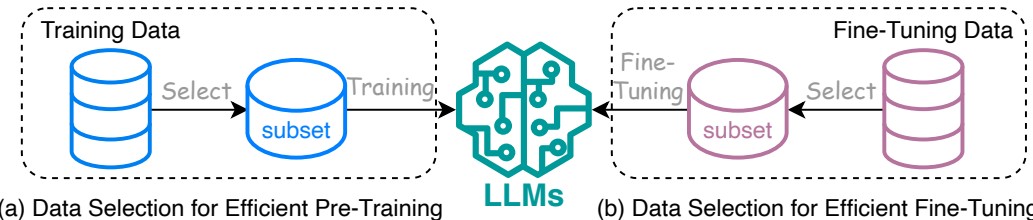

Figure 17: Illustrations of data selection techniques for LLMs.

Transformers. Ren et al. (2023a) extend SSMs and propose a general modular activation mechanism named Sparse Modular Activation (SMA), which unifies MoE, adaptive computation, dynamic routing and sparse attention, and further applies SMA to develop a novel architecture, SeqBoat, to achieve state-of-the-art quality-efficiency trade-off.

**Other Sequential Models**. Some other architectures have been proposed to replace the Transformer layer. For instance, Receptance Weighted Key Value (RWKV) model (Peng et al., 2023b) combines the advantages of recurring neural networks (RNN) and Transformers. Such combination utilizes the effective parallelizable training feature of Transformers coupled with the efficient inference ability of RNNs, thereby effectively tackling the challenges associated with long sequence processing, outperforming Transformer-based models such as BLOOM and OPT. Poli et al. (2023) propose Hyena, a sub-quadratic alternative to the attention mechanism to mitigate the quadratic cost in long sequences. This operator includes two efficient sub-quadratic primitives: an implicit long convolution and multiplicative element-wise gating of the input. Hyena facilitates the development of larger, more efficient convolutional language models for long sequences and outperforms RWKV and GPT-Neo on SuperGLUE tasks (Wang et al., 2019). Lastly, MEGABYTE (YU et al., 2023) breaks down long sequences into fixed-sized patches akin to tokens, comprising a patch embedder for encoding, a global module acting as a large autoregressive Transformer for patch representations, and a local module for predicting bytes within a patch.

# 3 Data-Centric Methods

## 3.1 Data Selection

Data selection is a fundamental technique for enhancing efficiency. As summarized in Figure 16, in the context of LLMs, data selection techniques have been primarily used for enhancing the efficiency of pre-training and fine-tuning.

### 3.1.1 Data Selection for Efficient Pre-Training

Data selection enhances LLMs pre-training efficiency by strategically selecting informative and diverse data samples during training. For example, SSPT (Glass et al., 2020) is a pre-training task based on the principles of reading comprehension. It involves selecting answers from contextually relevant text passages, which has shown notable improvements in performance across various Machine Reading Comprehension benchmarks. Yao et al. (2022a) propose a meta-learning-based method for selecting linguistically informative sentences which significantly elevates the quality of machine-generated translations. Xie et al. (2023b) propose DSIR,

a data selection method based on importance resampling for both general-purpose and specialized LLMs. It calculates how important different pieces of data are within a simpler set of features and chooses data based on these importance calculations. Experimental results demonstrate that DSIR achieves similar performance to expert curation across eight different target distributions. In the context of pre-training general-domain models, DSIR outperforms random selection and heuristic filtering baselines by 2–2.5% on the GLUE benchmark. Different from DSIR, Xie et al. (2023a) design DoReMi to address the distribution shift between pre-training and downstream tasks, which is also a critical problem for training data selection.

### 3.1.2 Data Selection for Efficient Fine-Tuning

Data selection can also boost LLM fine-tuning efficiency given that only a curated subset of examples is employed to refine the model. For example, Ivison et al. (2023) propose to use a few unlabeled data samples to retrieve similar labeled ones from a larger multitask dataset, improving task-specific model training. This method outperforms standard multitask data sampling for fine-tuning and enhances few-shot fine-tuning, yielding an 2-23% relative improvement. With the success of instruction tuning, many studies start focusing on the selection of high-quality instruction data to fine-tune LLMs. For example, Instruction Mining (Cao et al., 2023) presents a linear evaluation method to assess data quality in instruction-following tasks. It highlights the importance of high-quality data, showing that models trained with Instruction Mining-curated datasets outperform those trained on generic datasets in 42.5% of the considered cases. TS-DShapley (Schoch et al., 2023) is introduced to address the computational challenges of applying Shapley-based data valuation to fine-tuning LLMs. It employs an efficient sampling-based method that aggregates Shapley values computed from subsets to evaluate the entire training set. Low Training Data Instruction Tuning (LTD Instruction Tuning) (Chen et al., 2023b) challenges the need for large datasets in fine-tuning, showing that less than 0.5% of the original dataset is able to effectively train task-specific models without compromising performance. This approach enables more resource-efficient practices in data-scarce environments, combining selective data strategies with tailored training protocols for optimal data efficiency. AlpaGasus (Chen et al., 2024b) is a model fine-tuned on a mere 9K high-quality data samples, which are meticulously filtered from a larger dataset of 52K. It outperforms the original model trained on the full dataset and reduces the fine-tuning time by 5.7x, demonstrating the power of high-quality data in instruction-fine-tuning. Lastly, LIMA (Zhou et al., 2023a) fine-tunes LLMs with a small, selected set of examples, showing strong performance and challenging the need for extensive tuning. It generalizes well to new tasks, matching or exceeding GPT-4 in 43% of the considered cases.

## 3.2 Prompt Engineering

Prompt engineering (Liu et al., 2023a) focuses on designing effective inputs (i.e., prompts) to guide LLMs in generating desired outputs. It enhances inference efficiency by tailoring the input prompts or queries to better suit the capabilities of a specific language model. When used for some simple tasks, such as semantic classification, prompt engineering can even substitute fine-tuning to achieve high accuracy (Liu et al., 2022a). As summarized in Figure 18, prompt engineering techniques can in general be grouped into few-shot prompting, prompt compression, and prompt generation.

### 3.2.1 Few-Shot Prompting

Few-shot prompting aims to provide a LLM with a limited set of examples, referred to as demonstrations, to steer its understanding to a task it is required to execute (Wei et al., 2022a). These demonstrations are selected from the training corpus based on their similarity to the test example, and the LLM is expected to use the knowledge gained from these similar demonstrations to make the correct prediction (Dong et al., 2023). Few-shot prompting provides an efficient mechanism to use LLM by guiding the LLM to perform a wide variety of tasks without the need for additional training or fine-tuning. Furthermore, an effective few-shot prompting approach can make the created prompt concise enough to allow LLMs to quickly adjust to the task in high accuracy with only a slight increase of extra context, thus significantly improving inference efficiency. As illustrated in Figure 19, few-shot prompting techniques can in general be grouped into demonstration organization and template formatting.

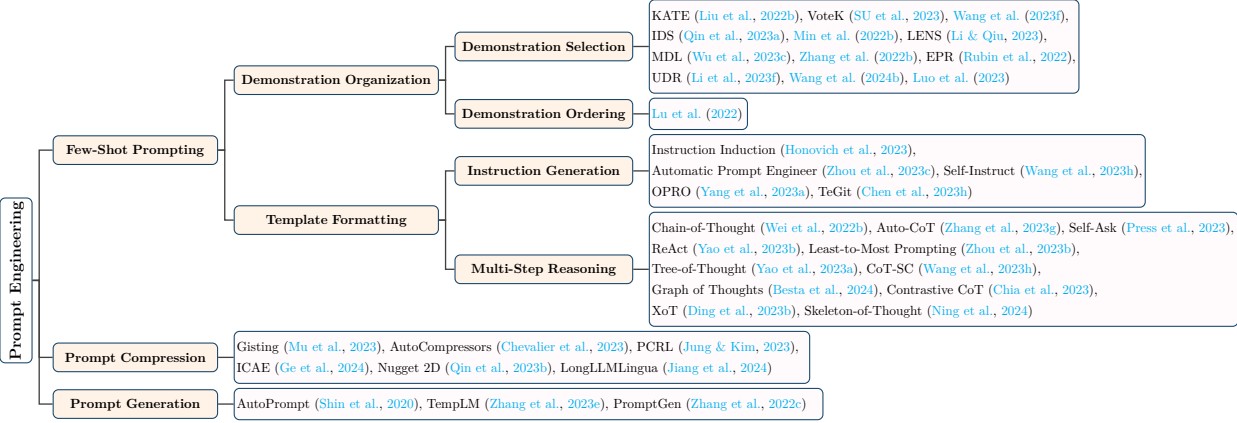

Figure 18: Summary of prompt engineering techniques for LLMs.

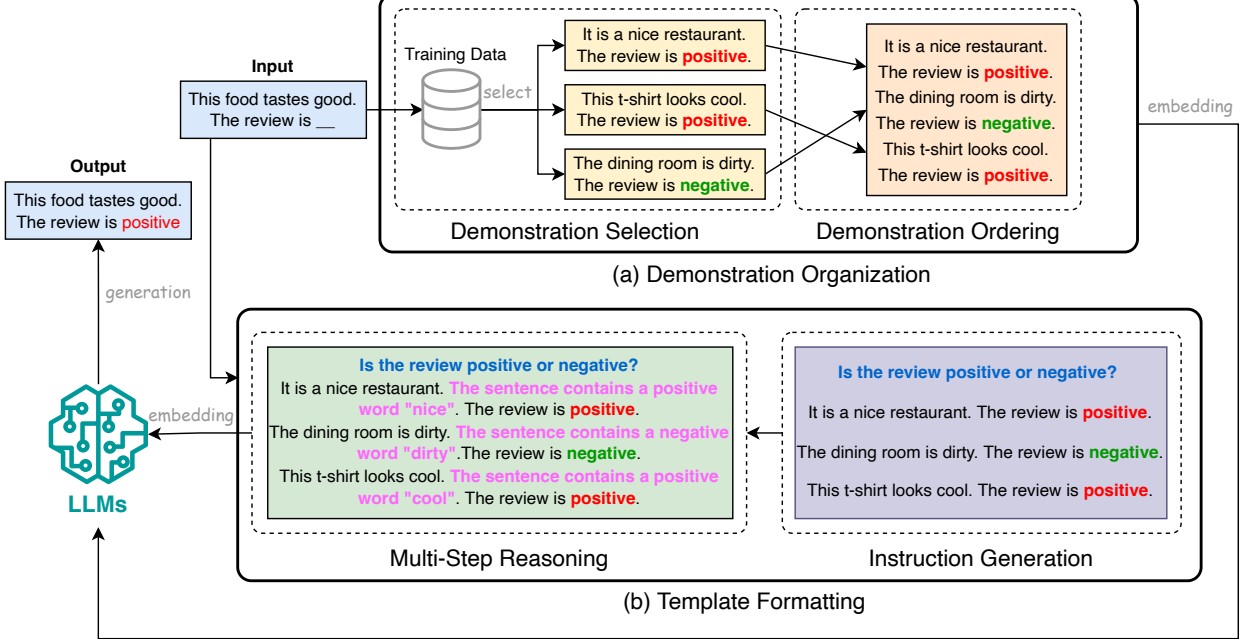

Figure 19: Illustrations of few-shot prompting techniques for LLMs.

**Demonstration Organization.** Demonstration organization refers to organizing the demonstrations in an appropriate way so as to form a suitable prompt for inference. Demonstration organization has a significant impact on the inference efficiency since improper organization may result in the processing of a considerable amount of unnecessary information, leading to significant slowdown. The optimization of demonstration organization comes from its two main steps: demonstration selection and demonstration ordering.

- *Demonstration Selection.* Demonstration selection aims to choose the good examples for few-shot prompting (Dong et al., 2023). In order to generate a satisfactory result, a good selection of demonstrations may only require a few number of demonstrations to be used for the prompt, thus making the prompt concise and straightforward for a more efficient inference. Existing demonstration selection techniques can be grouped into unsupervised methods (Liu et al., 2022b; SU et al., 2023; Wang et al., 2023f; Qin et al., 2023a; Min et al., 2022b; Li & Qiu, 2023; Wu et al., 2023c; Zhang et al., 2022b) and supervised methods (Rubin et al., 2022; Li et al., 2023f; Wang et al., 2024b; Luo

et al., 2023). Unsupervised methods aim to select the nearest examples from the training set using a predefined similarity function, such as L2 distance, cosine distance, and minimum description length (MDL) (Wu et al., 2023c). For example, KATE (Liu et al., 2022b) is an unsupervised selection method that directly uses the nearest neighbors of a given test sample as the corresponding demonstrations. VoteK (SU et al., 2023) is an improved version of KATE to resolve its limitation that requires a large set of examples to achieve good performance. Unlike KATE, VoteK increases the diversity of the demonstrations by penalizing examples similar to those already selected. In contrast, supervised methods require training a domain-specific retriever from the training set and using it for demonstration selection. For example, EPR (Rubin et al., 2022) is trained to select demonstrations from a small set of candidates initialized by the unsupervised retriever such as BM25 from the training corpus. UDR (Li et al., 2023f) further enhances EPR by adopting a unified demonstration retriever to unify the demonstration selection across different tasks. Compared to unsupervised methods, supervised methods often lead to a more satisfying generation result but require frequent adjustment of the retriever for handling the out-of-domain data, making them relatively less efficient for inference.

- *Demonstration Ordering.* After selecting representative samples from the training set, the next step is ordering these samples in the prompt. The order of the demonstrations also has a significant impact on the performance of the model. Therefore, selecting the right order of demonstrations can help the model quickly reach a good generation quality with fewer samples, thus improving the inference efficiency. To date, only a few studies have delved into this area. For example, Liu et al. (2022b) suggest arranging demonstrations based on their distance from the input, placing the closest demonstration furthest to the right. Lu et al. (2022) propose to develop both global and local entropy metrics and use the entropy metrics to set up the demonstration order.

**Template Formatting.** Template formatting aims to design a suitable template to form the prompt. A good template typically compiles all the information needed by LLMs into a brief statement, making the prompt and the entire input context as succinct as possible, thus leading to a higher inference efficiency. Template formatting can be divided into two parts: instruction generation and multi-step reasoning.

- *Instruction Generation.* The instruction of the template refers to a short description of the task. By adding instructions to the prompt, LLMs can quickly understand the context and the task they are currently performing, and thus may require fewer demonstrations to create a desirable prompt. The performance of a given task is highly affected by the quality of the instructions. The instructions vary not only between different datasets for the same task but also between different models. Unlike demonstrations that are usually included in traditional datasets, the generation of instructions is heavily dependent on human efforts. To enhance the efficiency of instruction generation, automatic instruction generation techniques have been proposed. For example, Instruction Induction (Honovich et al., 2023) and Automatic Prompt Engineer (Zhou et al., 2023c) demonstrate that LLMs can generate task instructions. Wang et al. (2023h) propose Self-Instruct, an approach that allows LLMs to align with self-generated instructions, highlighting their inherent adaptability. Experimental results show that it achieves an 33% improvement over the original model when applied on vanilla GPT3. Yang et al. (2023a) also discover that LLMs can be treated as an optimizer to iteratively generate better instructions for the target LLM and have applied this technique to various LLMs. Experiments demonstrate that the best prompts optimized by this method outperform human-designed prompts by up to 8% on GSM8K, and by up to 50% on Big-Bench Hard tasks. Chen et al. (2023h) develop TeGit for training language models as task designers, which can automatically generate inputs and outputs together with high-quality instructions to better filter the noise based on a given human-written text for fine-tuning LLMs. Despite the promise of automatic instruction generation methods, their complexity is still a major bottleneck for their real-world adoption.

- *Multi-Step Reasoning.* Multi-step reasoning (Huang & Chang, 2023) refers to techniques that guide the LLMs to produce a sequence of intermediate steps before outputting the final answer. Compared to fine-tuning, conducting specific task reasoning directly through this way is a more efficient approach. At the same time, its accuracy improvement is often not as good as fine-tuning. One

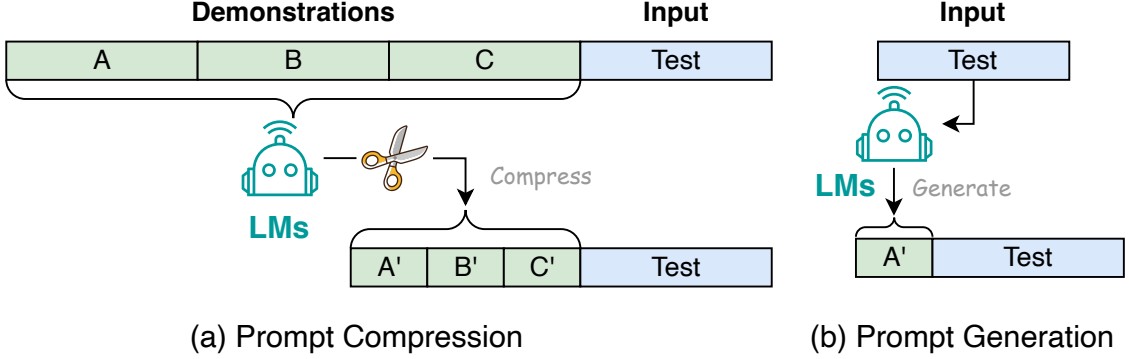

Figure 20: Illustrations of prompt compression (a) and prompt generation (b) for LLMs.

of the widely used techniques in multi-step reasoning is Chain-of-Thought (CoT) prompting (Wei et al., 2022b), which adds a series of reasoning steps into the prompt to make it more informative and comprehensive. Despite its advantages, it is still difficult for CoT to ensure the accuracy of every intermediate step (Dong et al., 2023). A number of techniques have been proposed to address this issue. For example, Auto-CoT (Zhang et al., 2023g) proposes to generate the CoT step by step from LLMs. Self-Ask (Press et al., 2023) incorporates the self-generated questions of each step into CoT. ReAct (Yao et al., 2023b) performs dynamic reasoning to create, maintain, and adjust high-level plans for acting, while interacting with external environments to incorporate additional information into reasoning. Least-to-Most Prompting (Zhou et al., 2023b) is a new milestone in CoT. It breaks down the complex question into smaller ones and answers them iteratively within the context of former questions and answers. Experimental results show that Least-to-Most Prompting can boost the accuracy of GPT-3 code-davinci-002 model with CoT prompting to 99.7% by using just 14 exemplars. Tree-of-Thought (ToT) (Yao et al., 2023a) expends CoT to include exploration over coherent units of text and deliberates decision-making processes. It outperforms GPT-4 with CoT prompting in the Game of 24 benchmark. CoT-SC (Wang et al., 2023h) introduces a novel decoding approach called self-consistency to replace the greedy decoding in CoT prompting. It starts by sampling various reasoning paths instead of just the greedy one and then determines the most consistent answer by considering all the sampled paths. Graph of Thoughts (GoT) (Besta et al., 2024) represents information produced by an LLM as a generic graph, with "LLM thoughts" as vertices and edges indicating dependencies between these vertices. Experimental results show that GoT increases the quality of sorting by 62% over ToT while reducing the cost by over 31%. Contrastive CoT (Chia et al., 2023) proposes to enhance language model reasoning by providing both valid and invalid reasoning demonstrations. XoT (Ding et al., 2023b) utilizes pretrained reinforcement learning and Monte Carlo Tree Search (MCTS) to integrate external domain knowledge into the thought processes of LLMs, thereby boosting their ability to efficiently generalize to new, unseen problems. Lastly, Skeleton of Thought (SoT) (Ning et al., 2024) proposes a method that first prompts the LLM to organize the output and then parallelizes the generation of different segments. Through splitting the serial generation into two different steps, it makes the generation workload more parallelizable. Therefore, it improves hardware utilization and provides speedups across LLM, revealing the possibility of exploiting data-level organization to enhance efficiency.

### 3.2.2 Prompt Compression

Prompt compression (Figure 20(a)) accelerates the processing of LLM inputs through either condensing lengthy prompt inputs or learning compact prompt representations. Mu et al. (2023) propose to train LLMs to distill prompts into a more concise set of tokens, referred to as gist tokens. These gist tokens encapsulate the knowledge of the original prompt and can be stored for future use. In doing so, it is able to compress prompts by up to 26x, leading to a reduction in floating-point operations per second (FLOPs) by up to

40%. Chevalier et al. (2023) propose AutoCompressors to condense long textual contexts into compact vectors, known as summary vectors, which can then be used as soft prompts for the language model. These summary vectors extend the model's context window, allowing it to handle longer documents with much less computational cost. In particular, AutoCompressors can utilize long contexts to improve perplexity of both fine-tuned OPT and LLaMA-2 on sequences of up to 30,720 tokens. Jung & Kim (2023) propose Prompt Compression with Reinforcement Learning (PCRL) that employs a policy network to directly edit prompts, aiming to reduce token count while preserving performance. It achieves an average reduction of 24.6% in token count across various instruction prompts. Ge et al. (2024) propose In-context Autoencoder (ICAE), which consists of a learnable encoder and a fixed decoder. The encoder compresses a long context into a limited number of memory slots, which the target language model can then condition on. With such design, ICAE is able to obtain 4x context compression. Nugget 2D (Qin et al., 2023b) represents the historical context as compact "nuggets" that are trained to enable reconstruction. Furthermore, it has the flexibility to be initialized using readily available models like LLaMA. Eperimental results show that Nugget 2D compresses context at a 20x compression ratio with a BLEU score of 98% for reconstruction, achieving nearly lossless encoding. Lastly, LongLLMLingua (Jiang et al., 2024) introduces a prompt compression technique containing question-aware coarse-to-fine compression, document reordering, dynamic compression ratios, and post-compression sub-sequence recovery to enhance LLMs' key information perception. Experimental results show that LongLLMLingua achieves 17.1% better performance over the original prompt with 4x fewer tokens as input to GPT-3.5-Turbo.

### 3.2.3 Prompt Generation

Prompt generation (Figure 20(b)) enhances efficiency by automatically creating effective prompts that guide the model in generating specific and relevant responses. For instance, AutoPrompt (Shin et al., 2020) proposes an automated method to generate prompts for a diverse set of tasks based on a gradient-guided search. It underscores the significance of human-written text in refining the quality and authenticity of data, emphasizing its pivotal role in optimizing LLM performance. Experimental results demonstrate that AutoPrompt outperforms manually created prompts on the LAMA benchmark in eliciting more precise factual knowledge from LLM. TempLM (Zhang et al., 2023e) proposes to combine generative and template-based methodologies to distill LLMs into template-based generators, offering a harmonized solution for data-to-text tasks. TempLM not only reduces the unfaithfulness rate of a fine-tuned BART model from 83% to 0%, but also substantially improves upon human-written ones in BERTScore. Lastly, PromptGen (Zhang et al., 2022c) considers dynamic prompt generation for knowledge probing based on pre-trained LLMs. It automatically generates prompts conditional on the input sentence and outperforms AutoPrompt on the LAMA benchmark.

## 4 LLM Frameworks

LLM frameworks can be in general grouped based on whether they support the tasks of training, fine-tuning, and inference. Specifically, frameworks that support training and/or fine-tuning aim to provide scalable, efficient, and flexible infrastructure that improves computation efficiency, reduces memory footprint, optimizes communication efficiency, and ensures reliability of the training/fine-tuning process. Frameworks that support inference focus on optimizing inference throughput and reducing memory footprint and latency. These frameworks offer a variety of deployment features to serve LLM requests. Table 2 provides a summary of existing LLM frameworks along with their key features.

**DeepSpeed.** Developed by Microsoft, DeepSpeed (Rasley et al., 2020) is an integrated framework for both training and serving LLMs. It has been used to train large models like Megatron-Turing NLG 530B (Smith et al., 2022) (in a joint effort with Nvidia Megatron framework) and BLOOM (Scao et al., 2023). Within this framework, DeepSpeed-Inference is the foundational library. A pivotal feature of DeepSpeed-Inference is ZeRO-Inference (Rajbhandari et al., 2020; 2021), an optimization technique created to address GPU memory constraints for large model inference. ZeRO-Inference distributes model states across multiple GPUs and CPUs, providing an approach to managing the memory constraints of GPUs.

Table 2: Comparison of LLM frameworks.

| Framework | Training | Fine-Tuning | Inference | Key Features |
|---|:---:|:---:|:---:|---|
| DeepSpeed | ✓ | ✓ | ✓ | 3D Parallelism with ZeRO (Rasley et al., 2020), ZeRO-2 (Rajbhandari et al., 2020), ZeRO Infinity (Rajbhandari et al., 2021) and ZeRO-Offload (Ren et al., 2021), Expert Parallelism (Rajbhandari et al., 2022), FlashAttention (Dao et al., 2022), PagedAttention (Kwon et al., 2023), Dynamic SplitFuse (Holmes et al., 2024), Continuous Batching (Yu et al., 2022), ZeroQuant (Wu et al., 2023b; Yao et al., 2023d), INT4 Quantization (Wu et al., 2023a), XTC (Ternary quantization) (Wu et al., 2022b), RLHF (Yao et al., 2023c), Kernel Optimizations, Diverse Hardware Support. |
| Megatron | ✓ | ✓ | ✓ | 3D Parallelism (Shoeybi et al., 2020; Narayanan et al., 2021), Sequence Paralellism (Korthikanti et al., 2023; Li et al., 2023d), Expert Parallelism (Singh et al., 2023), FasterTransformer (NVIDIA, 2023a), FlashAttention (Dao et al., 2022), Selective Activation Recomputation (Korthikanti et al., 2023). |
| Colossal-AI | ✓ | ✓ | ✓ | 3D Parallelism (Xu & You, 2023; Wang et al., 2023a; Bian et al., 2021), Sequence Parallelism (Li et al., 2023d), ZeRO Optimizer (Zhao et al., 2022), Auto-Parallelism (Liu et al., 2023c), Heterogeneous Memory Management (Fang et al., 2023), Expert Parallelism (Singh et al., 2023; Xue et al., 2023), PagedAttention (Kwon et al., 2023), FlashAttention-2 (Dao, 2024), Quantization (GPTQ (Frantar et al., 2023) and SmoothQuant (Xiao et al., 2023)), RLHF (Griffith et al., 2013; Singh et al., 2023). |
| Nanotron | ✓ | ✓ | ✓ | 3D Parallelism (Narayanan et al., 2021; Huang et al., 2019), Expert Parallelism (Singh et al., 2023), ZeRO Optimizer (Zhao et al., 2022), SSM (Gu & Dao, 2023) Support, Spectral μTransfer Parametrization (Yang et al., 2022), DoReMi (Xie et al., 2023a). |
| MegaBlocks | ✓ | ✓ | ✓ | FSDP (Zhao et al., 2023c), dropless-MoE (Gale et al., 2023), Integration with Megatron and vLLM. |
| FairScale | ✓ | ✓ | ✓ | FSDP (Zhao et al., 2023c), Pipeline Parallelism (Huang et al., 2019), AdaScale optimizer (Johnson et al., 2020). |
| Pax | ✓ | ✓ | ✓ | Data Parallelism, Tensor Parallelism (Shoeybi et al., 2020). |
| Composer | ✓ | ✓ | ✓ | FSDP (Zhao et al., 2023c), Elastic Sharded Checkpointing. |

Table 2: Comparison of LLM frameworks. (Continued)

| | | | | |
|---|---|---|---|---|
| OpenLLM | ❌ | ✅ | ✅ | FSDP (Zhao et al., 2023c), Quantization (GPTQ (Frantar et al., 2023), AWQ (Lin et al., 2023), SqueezeLLM (Kim et al., 2024), SpQR (Dettmers et al., 2024), LLM.int8 (Dettmers et al., 2022)), LangChain, Transformers Agents, Prometheus Metrics. |
| LLM Foundry | ❌ | ✅ | ✅ | FSDP (Zhao et al., 2023c), Continuous Batching (Yu et al., 2022). |
| vLLM | ❌ | ❌ | ✅ | Data Parallelism, Tensor Parallelism (Shoeybi et al., 2020), PagedAttention (Kwon et al., 2023), Continuous Batching (Yu et al., 2022), Quantization (GPTQ (Frantar et al., 2023), AWQ (Lin et al., 2023), SqueezeLLM (Kim et al., 2024)), Multi-LoRa (Wang et al., 2023g). |
| TensorRT-LLM | ❌ | ❌ | ✅ | 3D Parallelism, PagedAttention (Kwon et al., 2023), Continuous Batching (Yu et al., 2022), Quantization (GPTQ (Frantar et al., 2023), AWQ (Lin et al., 2023), SmoothQuant (Xiao et al., 2023)). |
| TGI | ❌ | ❌ | ✅ | Data Parallelism, Tensor Parallelism (Shoeybi et al., 2020), PagedAttention (Kwon et al., 2023), Continuous Batching (Yu et al., 2022), Quantization (BitsAndBytes Dettmers (2023), GPTQ (Frantar et al., 2023)). |
| RayLLM | ❌ | ❌ | ✅ | Data Parallelism, Tensor Parallelism (Shoeybi et al., 2020), Continuous Batching (Yu et al., 2022), Quantization (GPTQ (Frantar et al., 2023), AWQ (Lin et al., 2023), SqueezeLLM (Kim et al., 2024)), Prometheus Metrics. |
| MLC LLM | ❌ | ❌ | ✅ | TVM-based Compiler Acceleration (Feng et al., 2023; Chen et al., 2018), Continuous Batching (Yu et al., 2022), Quantized Model Support. |
| Sax | ❌ | ❌ | ✅ | Data Parallelism, Tensor Parallelism (Shoeybi et al., 2020), Serves Pax, JAX, and PyTorch models, Slice Serving, Prometheus Metrics. |
| Mosec | ❌ | ❌ | ✅ | Data Parallelism, Continuous Batching (Yu et al., 2022), Rust-based Task Coordinator, Prometheus Metrics. |

Another important feature of DeepSpeed-Inference is its deep fusion mechanism, which allows for the fusion of operations without the necessity for global synchronization by tiling computations across iteration space dimensions (Ren et al., 2021; Tang et al., 2021; Li et al., 2022; Lu et al., 2023). Building on this, the DeepSpeed Model Implementations for Inference (DeepSpeed MII) module introduces Dynamic Split-Fuse (Holmes et al., 2024), which leverages continuous batching (Yu et al., 2022) and non-contiguous KV caches to enable increased occupancy and higher responsivity for LLM serving. It also supports scaling beyond tensor, data, and pipeline parallelism (3D Parallelism) with Zero Infinity (Rajbhandari et al., 2021) and efficient post-training quantization to Int8 with ZeroQuant (Yao et al., 2022b), Int4 (W4A4) with Wu et al. (2023a) and ternary quantization with XTC (Wu et al., 2022b). Furthermore, the introduction of DeepSpeed-Chat (Yao et al., 2023c) adds chat support to the framework. This module focuses on training chatbot models across different scales, integrating techniques like Reinforcement Learning from Human Feedback (RLHF) (Griffith et al., 2013) with the DeepSpeed training system. Notably, its integration of the

ZeRO-Offload optimizer (Ren et al., 2021) facilitates training on both CPUs and GPUs, irrespective of their memory capacities.

**Megatron.** Megatron (Shoeybi et al., 2020) is Nvidia's efforts to streamline training and serving of LLMs such as GPT (Radford et al., 2019) and T5 (Raffel et al., 2020). It is the underlying framework used for Nvidia Megatron models (Shoeybi et al., 2020; Narayanan et al., 2021; Korthikanti et al., 2023). Central to Megatron's design is the strategic decomposition of the model's tensor operations, distributed across multiple GPUs, to optimize both processing speed and memory utilization, thus enhancing training throughput without compromising model quality (Shoeybi et al., 2020). In conjunction with 3D Parallelism, it also implements sequence parallelism and selective activation recomputation (Korthikanti et al., 2023), which enhances training efficiency. Megatron also uses FasterTransformer (NVIDIA, 2023a) for optimizing the inference process for large Transformer models and handling varying precision modes like FP16 and INT8, catering to diverse operational needs.

**Colossal-AI.** Colossal-AI (Li et al., 2023c) is a framework mainly designed for large-scale distributed training (Wang et al., 2023a). Colossal-AI unifies a wide range of parallelism techniques (Xu & You, 2023; Wang et al., 2023a; Bian et al., 2021) including sequence parallelism (Li et al., 2023d), auto-parallelism (Liu et al., 2023c), and Zero Redundancy Optimizer (Zhao et al., 2022). It also implements heterogeneous memory management (Fang et al., 2023) through a streamlined API. This integrated approach mitigates the steep learning curve often associated with orchestrating large-scale training in distributed environments. In addition, the framework integrates several other features like quantization (Frantar et al., 2023; Xiao et al., 2023), RLHF (Griffith et al., 2013), OpenMoE, and mixed-precision training.

**Nanotron.** Nanotron (HuggingFace, 2023a), introduced by Huggingface, is an LLM training framework with a primary focus on providing functionality with minimal overhead. As such, the library only subjectively incorporates the best optimizations required for modern LLM training requirements. Some highlights are its implementation of tensor, data, and pipeline parallelism with one-forward-one-backward pipeline engine, ZeRO-1 optimizer (Zhao et al., 2022), FP32 gradient accumulation, parameter tying/sharding and spectral µTransfer parametrization (Yang et al., 2022) for scaling up neural networks. Nanotron also incorporates DoReMi (Xie et al., 2023a) to further speed up training.

**MegaBlocks.** Developed by Databricks, MegaBlocks (Gale et al., 2023) is an LLM training framework for training Mixture-of-Experts (MoE) models. The core of MegaBlocks is dropless-MoE, a reformulation of MoE in terms of block-sparse operations, that avoids token dropping without sacrificing hardware efficiency. This design simplifies and accelerates training as it does not require the capacity factor as a hyper-parameter.

**FairScale.** Developed by Meta, FairScale (FairScale authors, 2021) is an extension library to PyTorch, dedicated to high-performance and large-scale training. As a highlight, FairScale uses Fully Sharded Data Parallel (FSDP) (Zhao et al., 2023c) as the preferred method for scaling the training operations of large neural networks. It uses AdaScale optimizer (Johnson et al., 2020) as its distributed optimizer.

**Pax.** Developed by Google, Pax (Google, 2023a) is a JAX-based distributed training framework. Pax has been used to train PaLM-2 (Anil et al., 2023) and Bard (Hsiao et al., 2023). It targets scalability and has reference examples for large model training, including across modalities (e.g., text, vision, speech). It supports data and tensor parallelism, and is heavily integrated with JAX and uses many libraries in the JAX ecosystem. Several key components Pax contains include SeqIO to handle sequential data processing, Optax for optimization, Fiddle for configuration, Orbax for checkpointing, PyGLove for automatic differentiation, and Flax for creating high-performance neural networks.

**Composer.** Composer (MosaicML, 2023a) is an LLM framework designed by Mosaic ML. It has been used to train Mosaic ML's MPT 7B and MPT 30B models and Replit's Code V-1.5 3B. The framework is built on top of PyTorch and provides a collection of acceleration methods that users can incorporate into their training loops or use with the Composer trainer. Composer supports FSDP for efficient parallelism, elastic shared checkpointing for robust intermittent training, and a dataset streaming implementation allowing the download of datasets from cloud blob storage on the fly during training. Composer also provides a functional API for integrating methods directly into its training loops, as well as a Trainer API which automatically implements a PyTorch-based training loop, reducing the workload for ML developers.

**OpenLLM.** OpenLLM (Pham et al., 2023) was made by BentoML to serve and fine-tune LLMs within production environments. Anchored within the BentoML ecosystem, OpenLLM makes it easy to self-host LLMs and integrate them with other cloud services. OpenLLM emphasizes on flexibility, SOTA LLM support, and streamlined APIs for self-hosting. Recognizing the diverse needs of production environments, OpenLLM supports the automatic generation of docker images. It also supports serving models as serverless endpoints using BentoML's cloud platform. OpenLLM further integrates advanced quantization techniques (GPTQ (Frantar et al., 2023), AWQ (Lin et al., 2023), SqueezeLLM (Kim et al., 2024), SpQR (Dettmers et al., 2024), LLM.int8 (Dettmers et al., 2022)) for efficient inference. OpenLLM's design also incorporates robust monitoring with Prometheus metrics and logging tools, ensuring that operational insights are readily available for performance tuning and troubleshooting.

**LLM Foundry.** LLM Foundry (MosaicML, 2023b) is a library developed by MosaicML for fine-tuning, evaluating, and serving LLMs. It supports distributed inference via FSDP and continuous batching (Yu et al., 2022) for efficient serving. It also has cloud integration with the MosaicML platform.

**vLLM.** vLLM (Kwon et al., 2023) is an open-source library for LLM inference and serving. It adopts a different design in how KV cache is stored in memory. Central to this design is PagedAttention, a mechanism that segments the attention key and value (KV) cache for a set number of tokens. Unlike contiguous space storage, PagedAttention's blocks for the KV cache are stored flexibly, similar to the virtual memory management in operating systems. This facilitates memory sharing at a block level across various sequences tied to the same request or even different requests, thus enhancing memory management efficiency in handling attention mechanisms. Hence, vLLM can reduce the total memory usage when using complex decoding techniques such as parallel sampling and beam search as the memory blocks can be shared across different candidate samples. It also allows on-demand buffer allocation, while also eliminating external fragmentation as the blocks are uniformly sized. Furthermore, vLLM incorporates safeguards that prevent GPU memory overflow due to the increasing size of KV cache by evicting and recovering blocks as needed via swapping and recomputation. vLLM also supports Multi-LoRA (Wang et al., 2023g), continuous batching (Yu et al., 2022) and quantization (GPTQ (Frantar et al., 2023), AWQ (Lin et al., 2023) and SqueezeLLM (Kim et al., 2024)). Lastly, it implements efficient kernels for both Nvidia and AMD GPUs.

**TensorRT-LLM.** TensorRT-LLM (NVIDIA, 2023b) is a streamlined library to serve LLMs. Built on top of the TensorRT engine, TensorRT-LLM integrates optimized kernels from FasterTransformer (Timonin et al., 2022) and employs tensor parallelism, facilitating efficient inference at scale across multiple GPUs and servers without necessitating developer intervention or model changes. It also integrates seamlessly with the Nvidia Triton Inference Server for serving LLMs. Additionally, it offers features like tensor parallelism, continuous batching (Yu et al., 2022), and PagedAttention (Kwon et al., 2023) as well as supports a wide range of quantization modes and techniques.

**Text-Generation-Inference (TGI).** Text-Generation-Inference (TGI) (HuggingFace, 2023b) is a high-performance LLM serving library developed by Huggingface and is used to power Hugging Chat. TGI supports a large variety of LLMs, and offers a wide range of features like tensor parallelism, continuous batching (Yu et al., 2022), efficient attention mechanisms like FlashAttention (Dao et al., 2022) and PagedAttention (Kwon et al., 2023), and quantization (BitsAndBytes (Dettmers, 2023), GPTQ (Frantar et al., 2023)) support.

**RayLLM.** RayLLM (Ray Project, 2023) is an LLM serving framework as part of the Ray ecosystem (Moritz et al., 2018). Built with Ray Serve and the Ray library developed by AnyScale, RayLLM eases LLM serving in multi-GPU and multi-node systems. At the core of RayLLM is the leveraging of Ray's inherent distributed computing capabilities. RayLLM integrates Ray's distributed task scheduling and execution mechanisms, ensuring that LLM tasks are efficiently distributed across available resources. RayLLM also simplifies adding new LLMs for custom use cases, and offers auto-scaling support and high-performance features like continuous batching (Yu et al., 2022), quantization (GPTQ (Frantar et al., 2023), AWQ (Lin et al., 2023), SqueezeLLM (Kim et al., 2024)), and monitoring via Prometheus metrics. It also comes with advanced monitoring support as well which includes a CLI and a web frontend that can be used to compare and rank models and get cost and latency estimates.

**MLC LLM.** MLC LLM (Machine Learning Compilation for Large Language Models) (MLC-LLM, 2023) allows individuals to develop, optimize, and deploy LLMs on a wide range of platforms such as mobile phones and web browsers. Central to MLC LLM is its focus on machine learning compilation techniques. MLC-LLM compiles LLMs and deploys them in a process that is inherently tailored to the specific capabilities and constraints of each platform and hardware (Chen et al., 2018; Shao et al., 2022; Feng et al., 2023). This platform-native approach ensures that LLMs are not only efficient but also highly optimized for the platforms in which they operate.

**Sax.** Sax (Google, 2023b) is a platform designed by Google for serving Pax, JAX, and PyTorch models for inference tasks. It supports distributed inference with tensor and data parallelism. It also integrates easily with Google Cloud and cloud monitoring with Prometheus metrics.

**Mosec.** Mosec (Yang et al., 2021) is a serving framework built to streamline the serving of machine learning models into backend services and microservices. Mosec's key features include continuous batching (Yu et al., 2022), pipelined stages for handling mixed workloads, and essential cloud features like model warmup, graceful shutdown, and Prometheus monitoring metrics, making it easily manageable by Kubernetes or other container systems.

## 5 Concluding Remarks

In this survey, we provide a systematic review of efficient LLMs, an important area of research aimed at democratizing LLMs. We start with motivating the necessity for efficient LLMs. Guided by a taxonomy, we review algorithm-level and system-level efficient techniques for LLMs from model-centric and data-centric perspectives respectively. Furthermore, we review LLM frameworks with specific optimizations and features crucial for efficient LLMs. We believe that efficiency will play an increasingly important role in LLMs and LLMs-oriented systems. We hope this survey could enable researchers and practitioners to quickly get started in this field and act as a catalyst to inspire new research on efficient LLMs.

## 6 Acknowledgement

We would like to thank the action editor Greg Durrett and anonymous reviewers of Transactions on Machine Learning Research for their helpful and constructive comments.

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
