# OpenReview forum: "Efficient Large Language Models: A Survey"
_TMLR — Accepted by TMLR_

### Review · Reviewer_h5tv · 2024-02-12

**Summary Of Contributions:**

This is a survey paper on efficient LLMs. The overall structure of the survey is shown in Figure 3 – the survey is into three major sections: model-centric methods (model compression, efficient pre-training, efficient fine-tuning, efficient inference, efficient architecture), data-centric methods (data selection and prompt engineering), and frameworks (e.g., DeepSpeed, Megatron, etc.)

**Audience:**

Yes

**Claims And Evidence:**

Yes

**Requested Changes:**

Please address every point in the "weaknesses and comments" section above if the authors think they make sense.

**Strengths And Weaknesses:**

Strengths

The authors claim that they will actively maintain the repo of the paper list in each section; that will be fantastic.

I have worked on many of the fields in the paper, but not all. I have learned a lot from this survey, and I believe it will be beneficial for the community.

The figures throughout the survey paper (that describe the difference among different groups of methods (e.g., Figure 5, Figure 7, Figure 9, and many others) are very helpful.


---

Weaknesses and other comments

One complaint I have is that many sections (especially the ones I’m not familiar with) read like a list of paper summaries. There is no way for me to tell which method I should be using at the end. Is one method better than the other? Which methods are mostly used in today’s products (and most useful for industry practitioners)? Which methods are most promising for researchers to dig into? On the other hand, which methods are already abandoned or outdated? It’s unclear what my takeaway is after reading some of the sections.
- There are many examples throughout the paper. One example is quantization on Page 5. Many methods are enumerated. Which one should practitioners and researchers use? Which ones are already out-of-date?
- Another example is in Section 2.5 (efficient architecture design); in the paragraph “feature information reduction,” I’m wondering if anyone is still using Funnel transformer or Nystromformer – which ones are outdated, and which ones are still promising today (so that we should keep investigating)?
- Table 2 in Page 30 (comparing different LLM frameworks), on the other hand, has a clear takeaway and I really appreciate the table

(Personal preference:) I would appreciate it more if the authors highlight the mostly used and most promising approaches in each section. The details should be at least a paragraph and hopefully contain an equation or a figure when necessary.
- For example, FlashAttention and FlashAttention-2 are widely used, but the description isn’t detailed enough especially for people who don't know much about hardware / FlashAttention. Hopefully there could be a figure and a discussion on backward pass and how it saves time and memory.

For quantization (Section 2.1), I would appreciate a discussion on the tradeoff between compression rate and quality.

In the table detailing pretraining costs (Section 2.1), I’m a bit confused why Llama-2 is in a different unit compared to others.

In efficient fine-tuning (e.g., Section 2.3.1), the line of work on hypertuning (https://arxiv.org/abs/2211.12485) could be relevant.

In efficient architecture (Section 2.5), some of the other subsections also contribute to building today’s long context LLMs. In the long context LLM section, perhaps it’s prudent to mention that some other sections’ approaches are also relevant (e.g., FlashAttention).

In long-context LLM (Section 2.5.3), I wonder if Memwalker (https://arxiv.org/abs/2310.05029), RAPTOR (https://arxiv.org/abs/2401.18059) are relevant, as well as other recent methods that traverse a memory tree / memory document. More recently (I understand this just came out), gist memory: https://arxiv.org/pdf/2402.09727.pdf.

I wonder if it’s worth referring to other survey papers in Section 3.2.1 (related to prompting).

When I was clicking into different papers in the bibliography, I found that the format isn’t consistent. Some papers use initials for author names, but most papers use full names. Some papers have capitalized author names. Different arXiv papers’ citation formats are also very different. Many papers only include the author names and titles without arXiv / conference proceeding reference.

---

> ### Author Response · Authors · 2024-03-18
> **Response to Reviewer h5tv**
>
> Thank you for all the constructive suggestions.
>
> > One complaint I have is that many sections (especially the ones I’m not familiar with) read like a list of paper summaries. There is no way for me to tell which method I should be using at the end.
>
> In our survey, the selection of papers is organized based on the chronological order of their publication. Our investigation reveals that the most recent publications often achieve state-of-the-art (SOTA) performance. This arrangement facilitates readers' understanding of the latest developments in the subfield.
>
> > Is one method better than the other? Which methods are mostly used in today’s products (and most useful for industry practitioners)? Which methods are most promising for researchers to dig into? On the other hand, which methods are already abandoned or outdated? It’s unclear what my takeaway is after reading some of the sections.
>
> We have made amendments to our manuscript in accordance with the issues and suggestions you raised, as indicated by the newly added blue text in each section. Previously, our survey organized the discussed papers by their publication date. Following your feedback, we have implemented the following modifications: **1**. We have delineated the relationships between earlier and later methods, and **2**. If there is a comparative relationship between two methods, we now clarify how the more recent approaches outperform its predecessors in specific tasks and metrics.
>
> With these newly incorporated modifications, we believe readers will gain a clear understanding of which methods are the most cutting-edge, which comparative baselines merit further exploration, and the problems present in earlier approaches that subsequent methods have addressed.
>
> > There are many examples throughout the paper. But one example is quantization on Page 5. Many methods are enumerated. Which one should practitioners and researchers use? Which ones are already out-of-date?
>
> In the quantization section, we have summarized the methodologies in chronological order and additionally introduced orthogonal or comparative relationships between them. The newly revised sections aim to inform readers which methodologies are currently SOTA and which are considered outdated.  **Please refer to the newly added blue text in Section 2.1.1 of our manuscript for further details**.
>
> > Another example is in Section 2.5 (efficient architecture design); in the paragraph “feature information reduction,” I’m wondering if anyone is still using Funnel transformer or Nystromformer – which ones are outdated, and which ones are still promising today (so that we should keep investigating)?
>
> Compared to other attention optimization strategies, recent works related to feature information reduction are less common, possibly because the methods are considered outdated. Therefore, we have decided to remove this section from our paper.
>
> >  (Personal preference:) I would appreciate it more if the authors highlight the mostly used and most promising approaches in each section. The details should be at least a paragraph and hopefully contain an equation or a figure when necessary.
>
> >For example, FlashAttention and FlashAttention-2 are widely used, but the description isn’t detailed enough especially for people who don't know much about hardware / FlashAttention. Hopefully there could be a figure and a discussion on backward pass and how it saves time and memory.
>
>
> We have emphasized the most classic (i.e., most commonly used) algorithms in most of the sections in our paper. Furthermore, in response to your suggestion for a detailed explanation of FlashAttention and FlashAttention-2, we have included their illustrations in the paper. Please refer to section **2.5.1 Hardware-Assisted Attention” part for further details**.
>
> > For quantization (Section 2.1), I would appreciate a discussion on the tradeoff between compression rate and quality.
>
> We have added a discussion part for quantization (Section 2.1), focusing on the trade-off between quantization compression rates and the quality of model performance. **Please refer to section 2.1 discussion part for further details**.
>
> > In efficient fine-tuning (e.g., Section 2.3.1), the line of work on hypertuning (https://arxiv.org/abs/2211.12485) could be relevant.
>
> We have incorporated a discussion regarding this paper into our manuscript. **Please refer to section 2.3.1 prefix tuning part for further details**.
>
> > “In efficient architecture (Section 2.5), some of the other subsections also contribute to building today’s long context LLMs. In the long context LLM section, perhaps it’s prudent to mention that some other sections’ approaches are also relevant (e.g., FlashAttention).
>
> We have added a discussion about FlashAttention in the Long Contexts LLMs chapter. **Please review the modified content in the first paragraph of section 2.3.1, highlighted in blue, in the paper**.

---

> > ### Author Response · Authors · 2024-03-18
> > **Response to Reviewer h5tv Part II**
> >
> > > In long-context LLM (Section 2.5.3), I wonder if Memwalker (https://arxiv.org/abs/2310.05029), RAPTOR (https://arxiv.org/abs/2401.18059) are relevant, as well as other recent methods that traverse a memory tree / memory document.
> >
> > We have incorporated Memwalker into our paper. **Please refer to section 2.5.3, "Segmentation and Sliding Window," for further details**.
> >
> > > I wonder if it’s worth referring to other survey papers in Section 3.2.1 (related to prompting).
> >
> > Following your suggestion, we have added references to papers on prompting, such as "Towards Reasoning in Large Language Models: A Survey," in the Multi-Step Reasoning part of Section 3.2.1.
> >
> > > When I was clicking into different papers in the bibliography, I found that the format isn’t consistent. Some papers use initials for author names, but most papers use full names. Some papers have capitalized author names. Different arXiv papers’ citation formats are also very different. Many papers only include the author names and titles without arXiv / conference proceeding reference.
> >
> > Thank you for your valuable feedback. We have revised and standardized the format of the references accordingly.

---

> > ### Comment · Reviewer_h5tv · 2024-04-03
> > **Thank you for the response and the update**
> >
> > The authors have updated the manuscript quite a bit in blue text, and I really appreciate that. I skimmed through most of the additions and I agree that I'd have a better sense of what's the most updated methods that are worth investigating or implementing into the real world. As I discussed before I personally learned a lot by reading through the survey and the sections are nicely organized, so I think this will benefit the community.

---

> > > ### Author Response · Authors · 2024-04-07
> > > **Thank you!**
> > >
> > > We want to thank the reviewer again for the positive feedback and the very kind words. We as the authors really appreciate it.

---

### Review · Reviewer_npES · 2024-02-18

**Summary Of Contributions:**

This paper presents a very comprehensive survey on the efficiency of LLMs. Different from existing efficient LLMs surveys that typically focus on one specific aspect (e.g. inference efficiency), the concept of efficiency here is broad, covering model-centric efficiency such as model compression, efficient pretraining, efficient fine-tuning, efficient inference, and efficient architecture, data-centric efficiency that includes data selection and prompt engineering, and frameworks that compares different open-source libraries. Each of these subcategories are further classified with many details. I appreciate that the authors constructed such a taxonomy and put different types of efficiency into a single survey.

**Audience:**

Yes

**Claims And Evidence:**

Yes

**Requested Changes:**

As mentioned in the weaknesses above, I think more discussions on the connections and pros/cons of different research lines inside each subtopic should be presented to provide more insights and make the paper coherent. This will offer a clearer picture on what is going on in this regime as well. Currently the paper is merely describing each paper under the constructed taxonomy, with few discussions and insights.

**Strengths And Weaknesses:**

### Strengths

1. Efficiency is one of the most important topics in LLMs, and there are many research papers out every week focusing on different types of efficiencies. This survey is very comprehensive to put various efficiencies together. It is a nice read to understand the big picture of these research works.
2. Inside each of the sub-topics such as model compression and efficient fine-tuning, the authors did a good job to further construct the taxonomy, and each subsection in this paper is like a nice separate survey on its own.

### Weaknesses

1. Currently the survey reads more like a bland description of each research work inside each subtopic. However, I think it is important to summarize and compare different lines of research that aim for the same goal to understand the connections among different research lines as well as their pros/cons. For example, in model compression, there are different types of approaches such as quantization, pruning, and knowledge distillation. What are the pros/cons of each of these methods? What is the current preferred practice? What is the SOTA for model compression? Why are some of the lines more popular than the others? What is recommended to compress a model and why? Discussing these questions is important for the readers to understand the different tradeoffs and properties from these methods.
2. Another example aligning with the first point is on parameter-efficient fine-tuning. It is not wise to group LoRA separately from the adapters, when LoRA is more like a special case of adapters and LoRA is also often referred to as LoRA adapters. Similar to before, among these many different PEFT methods, what are their connections? Why does LoRA stand out as the most popular one? What are their pros and cons? For example, [1] presents a unified view to connect these different methods – I would like to see this kind of discussion for each subsection of the paper to raise more interesting insights and comparisons, rather than only blandly describing a lot of papers.

[1] He et al. Towards a Unified View of Parameter-Efficient Transfer Learning. ICLR 2022.

---

> ### Author Response · Authors · 2024-03-18
> **Response to Reviewer npES**
>
> > Currently the survey reads more like a bland description of each research work inside each subtopic. However, I think it is important to summarize and compare different lines of research that aim for the same goal to understand the connections among different research lines as well as their pros/cons. For example, in model compression, there are different types of approaches such as quantization, pruning, and knowledge distillation. What are the pros/cons of each of these methods? What is the current preferred practice? What is the SOTA for model compression? Why are some of the lines more popular than the others? What is recommended to compress a model and why? Discussing these questions is important for the readers to understand the different tradeoffs and properties from these methods.
>
> Thank you for all the constructive suggestions.
>
> +  We have added definitions, limitations/settings of use, and specific categories of efficiency (parameters, training, inference, serving, data) for each sub-method within their sections, providing readers with insights into the pros/cons of each type of method.
>
> + In each section, we have introduced connections among the methods, such as whether they are orthogonal or iterative improvements to other methods.
>
> + We organized the methods according to their publication date and compared their performance under the benchmarks mentioned in the paper, contrasting current methods with previous ones. This aids readers in understanding the current state-of-the-art (SOTA) models and the most popular models.
>
>  **Please refer to the newly added blue text in each subsection.**
>
> > Another example aligning with the first point is on parameter-efficient fine-tuning. It is not wise to group LoRA separately from the adapters, when LoRA is more like a special case of adapters and LoRA is also often referred to as LoRA adapters. Similar to before, among these many different PEFT methods, what are their connections? Why does LoRA stand out as the most popular one? What are their pros and cons? For example, [1] presents a unified view to connect these different methods – I would like to see this kind of discussion for each subsection of the paper to raise more interesting insights and comparisons, rather than only blandly describing a lot of papers.
>
> The main goal of our survey is to categorize methods as precisely as possible, and people use those names to specifically refer to those specific ways of inserting extra parameters. Therefore, we discuss Adapter-based tuning and LoRA separately and introduce the specific definitions of methods in the first sentence of each subsection to tell the difference.Specifically, Adapters are bottleneck-like trainable modules integrated into LLMs, which first down-project the input feature vector followed by a non-linear layer and then up-project back to the original size. Differently, LoRA is often referred to using the low-rank decomposition strategy to replace the gradient change. And Prefix-Tuning adds a series of trainable vectors, known as prefix tokens, to each layer in an LLM.

---

> > ### Comment · Reviewer_npES · 2024-04-08
> >
> > Thank you for the response and the update of the paper! I read the updated paper and I feel the comparison of different methods are still not emphasized enough. It may be better to have a separate paragraph or subsection to discuss these methods in a more holistic tone, for example, Section 2.1 is about model compression and the subsequent subsections are different types of methods for model compression. At the end of Section 2.1, it is worth having a separate subsection to summarize these methods in from holistic perspective -- and this can be done for every category basically. These discussions could be a unique thing of this paper.

---

### Review · Reviewer_28rG · 2024-03-04

**Summary Of Contributions:**

This paper gives as a comprehensive literature review delving into the various methodologies employed to enhance the efficiency of Large Language Models (LLMs). Its primary objective is to serve as a valuable repository of knowledge for both researchers and practitioners, providing a comprehensive understanding of the latest advancements and potentially inspiring innovative contributions. The techniques discussed in this paper are systematically categorized into three primary focus areas: model-centric, data-centric, and framework-centric.

**Audience:**

Yes

**Claims And Evidence:**

Yes

**Requested Changes:**

Please address concerns in the above section.

**Strengths And Weaknesses:**

This is a very timely survey in that the importance of efficient LLMs keeps increasing as the size of LLMs and their computational cost become bigger and bigger. It includes comprehensive lists of different directions and corresponding representative papers.

Efficiency is often a broad term since it could be one of model size, training/inference time/memory, and many others. Some methods only target a single metric, and others can benefit multiple aspects. Therefore, clearly demonstrating what efficiency criteria each approach improves will be helpful, especially for people who have their own specific optimization requirements to meet the computational budget.

Given that there have been many studies on efficient language models since a few decades ago (e.g., efficient transformers by Tay et al. is a good survey on BERT-era) and this paper is a survey on efficient *large* language models, it would be great to have a clear distinction on that. Of course, it is difficult to say how large a language model should be to be called an LLM. Also, many previous approaches for small language models could also apply to LLMs without or with a slight modification. Nevertheless, it would be necessary to make this survey different from the existing survey by narrowing down its scope.

Having an unbiased position in any particular direction looks objective, but giving some insights on the pros and cons (or limitations) of each direction/method will be advantageous.
Moreover, the paper should give a deeper view of the relationship between different approaches. For example, some are seamlessly orthogonal, but some are not.

---

> ### Author Response · Authors · 2024-03-18
> **Response to Reviewer 28rG**
>
> ### 1. Response for Weaknesses:
>
> > Therefore, clearly demonstrating what efficiency criteria each approach improves will be helpful, especially for people who have their own specific optimization requirements to meet the computational budget.
>
> Thank you for all the constructive suggestions. In our revision, we emphasize the efficiency criteria a particular work improves. For example, in **section 2.1**, ‘System-Level Inference Efficiency Optimization’ part , we added: “Then, Orca employs iteration-level scheduling to decide batch sizes. When a sequence in a batch is completed, it is substituted with a new one, **resulting in improved GPU utilization compared to static batching, showing 36.9× throughput improvement at the same level of latency compared to FasterTransformer**”.
>
> > Given that there have been many studies on efficient language models since a few decades ago (e.g., efficient transformers by Tay et al. is a good survey on BERT-era) and this paper is a survey on efficient large language models, it would be great to have a clear distinction on that. Of course, it is difficult to say how large a language model should be to be called an LLM. Also, many previous approaches for small language models could also apply to LLMs without or with a slight modification. Nevertheless, it would be necessary to make this survey different from the existing survey by narrowing down its scope.
>
> The paper "Efficient Transformers: A Survey" by Tay et al. focuses on various Transformer models and their efficiency improvements, primarily in computation and memory aspects. It discusses different types of efficient Transformers, especially focusing on efficient attention design, such as those improving on the original architecture's quadratic complexity problem.
> While our survey deals specifically with the efficiency challenges and techniques of Large Language Models (with parameters exceeding 7 billion), covering a broader efficient spectrum that includes model-centric methods, data-centric methods, and specialized frameworks for LLMs. Our survey is more general and systematic for LLMs,  and  It focuses on the scalability and efficiency of models with billions of parameters, emphasizing the necessity for advancements due to their substantial resource demands.
>
> > Having an unbiased position in any particular direction looks objective, but giving some insights on the pros and cons (or limitations) of each direction/method will be advantageous.
>
> In the first paragraph of each subsection, we usually describe the settings or limitations under which these methods should be applied. For instance, in **section 2.1.4** Knowledge Distillation,  we define the White-box KD and black-box KD:
>
>  “White-box KD refers to KD techniques where the parameters or logits of the teacher LLM are used in the distillation process.”
> “Different from white-box KD, in black-box KD, only the outputs generated from the teacher LLM are used in the distillation process.”
>
> And then we give a summary of how these methods specifically improve efficiency in certain aspects.
>
> > Moreover, the paper should give a deeper view of the relationship between different approaches. For example, some are seamlessly orthogonal, but some are not.
>
> We have incorporated relationships among the methods within each category in the paragraphs, such as orthogonality, similarity, or contrast, and briefly mentioned the comparison with the baseline for each method. Please refer to the newly added blue text in each subsection.

---

### Decision · Action_Editor_egD4 · 2024-04-17

**Recommendation:** Accept as is

**Comment:**

The reviewers and I uniformly found this survey to be informative and timely. It serves as a great collection of papers, and if the repository is continually updated as promised, it will be great a resource that I imagine myself pointing students to in the future.

The main weakness of this paper is the lack of a stronger position on the pros and cons of different methods. The update of the paper does a good job of comparing approaches within a single section, particularly by reporting results to compare multiple approaches. However, there is less insight or guidance on which of the many research areas in the overall taxonomy are most worth pursuing, or which have yielded the strongest results thus far. There is a brief discussion added about quantization, for example, but it mostly says that there has been a lot of work, yet tradeoffs remain.

I see this as somewhat difficult to add within the current structure of the survey. The authors are encouraged to take a step in the direction of adding comparison where possible (reviewer npES particularly thought that such comparisons would be valuable), but I believe the paper is above the bar for survey papers as is.

**Audience:**

Yes. The reviewers and I found this survey to be informative and timely.

**Claims And Evidence:**

Yes. This is a survey paper, and it succeeds at surveying a range of concepts and methods related to efficiency in LLMs.

---

> ### Author Response · Authors · 2024-05-16
> **Camera Ready Submitted**
>
> Dear action editor and reviewers,
>
> We want to let you know that we have submitted the camera ready version of our survey. We would like to express our sincere gratitude to the action editor Greg Durrett and all the anonymous reviewers for your informative and constructive comments that helped us improve the quality of our survey. We took those comments seriously when we prepared the camera ready version. We really appreciate it.